# SUFFICIENT AND NECESSARY EXPLANATIONS (AND WHAT LIES IN BETWEEN)

## ABSTRACT

As complex machine learning models continue to be used in high-stakes decision settings, explaining their predictions is crucial. Post-hoc explanation methods aim to identify which features of an input $\mathbf{x}$ are important to a model's prediction $f(\mathbf{x})$. However, explanations often vary between methods and lack clarity, limiting the information we can draw from them. To address this, we formalize two precise concepts—*sufficiency* and *necessity*—to quantify how features contribute to a model's prediction. We demonstrate that, although intuitive and simple, these two types of explanations may fail to fully reveal which features a model considers important. To overcome this, we propose and study a unified notion of importance that spans the entire necessity-sufficiency axis. Our unified notion, we show, has strong ties to other popular notions of feature importance, like those based on conditional independence and game-theoretic quantities like Shapley values. Lastly, through various experiments, we demonstrate that generating explanations along the necessity-sufficiency axis can uncover important features that may otherwise be missed and reveal that many post-hoc methods only provide features that are sufficient rather than necessary.

## 1 INTRODUCTION

Over recent years, modern machine learning (ML) models, mostly deep learning-based, have achieved impressive results across several complex domains. Models can now solve difficult image classification, inpainting, and segmentation problems, perform accurate text and sentiment analysis, predict the three-dimensional conformation of proteins, and more (LeCun et al., 2015; Wang et al., 2023). Despite their success, the rapid integration of these models into society requires caution (The White House, 2023). Modern ML systems are black-boxes, comprised of millions of parameters and non-linearities that obscure their prediction-making mechanisms from everyone. This lack of clarity raises concerns about explainability, transparency, and accountability (Zednik, 2021; Tomsett et al., 2018). Thus, understanding how these models work is essential for their safe deployment.

The lack of explainability has spurred research efforts in eXplainable AI (XAI), with a major focus on developing post-hoc methods to explain black-box model predictions, especially at a *local* level. For a model $f$ and input $\mathbf{x} \in \mathbb{R}^d$, these methods aim to identify which features in $\mathbf{x}$ are *important* for the model's prediction, $f(\mathbf{x})$. They do so by estimating a notion of importance for each feature (or groups), which allows for a ranking of importance. Popular methods include CAM (Zhou et al., 2016), LIME (Ribeiro et al., 2016), gradient-based approaches (Selvaraju et al., 2017; Shrikumar et al., 2017; Jiang et al., 2021), rate-distortion techniques (Kolek et al., 2022), Shapley value-based explanations (Chen et al., 2018b; Teneggi et al., 2022; Mosca et al., 2022), perturbation-based methods (Fong & Vedaldi, 2017; Fong et al., 2019; Dabkowski & Gal, 2017), among others (Chen et al., 2018a; Yoon et al., 2018; Jethani et al., 2021; Wang et al., 2021; Ribeiro et al., 2018). However, many of these approaches lack rigor, as the meaning of their computed scores is often ambiguous. For example, it's not always clear what large or negative gradients signify or what high Shapley values reveal about feature importance. To address these concerns, other research has focused on developing explanation methods based on logic-based definitions (Ignatiev et al., 2020; Darwiche & Hirth, 2020; Darwiche & Ji, 2022; Shih et al., 2018), conditional hypothesis testing Teneggi et al. (2023); Tansey et al. (2022), among formal notions. While these methods are a step towards rigor, they have drawbacks, including reliance on complex automated reasoners and limited ability to communicate their results in an understandable way for human decision-makers.

In this work, we advance XAI research by providing formal mathematical definitions of *sufficient* and *necessary* features for explaining complex ML models. First, we illustrate how, although informative, sufficient and necessary explanations offer incomplete insights into feature importance. To address this, we propose and study a more general unified framework for explaining models. Finally, we offer two novel perspectives on our framework through the lens of conditional independence and Shapley values, and crucially, show how it reveals new insights into feature importance.

## 1.1 Summary of our Contributions

We propose and study two approaches, sufficiency, and necessity, which evaluate the contribution of a set of features in $\mathbf{x}$ toward a model prediction $f(\mathbf{x})$. A sufficient set preserves the model's output, while a necessary set, when removed, renders the output uninformative. Although the two concepts appear complementary, their precise relationship remains unclear. How similar are sufficient and necessary subsets? How different? To address these questions, we study the two concepts and propose a *unification* of both. Our contributions are summarized as follows:

1. We formalize precise mathematical definitions of sufficient and necessary features for model predictions that are related but complementary to those in previous works.

2. We propose a unified approach that combines sufficiency and necessity, exploring when and how they align or differ. Additionally, we motivate its utility by highlighting its connections to conditional independence and Shapley values, a game-theoretic measure of feature importance.

3. Through experiments of increasing complexity, we demonstrate how a unified perspective uncovers new, significant, and more comprehensive insights into feature importance.

## 2 Sufficiency and Necessity

**Notation & Setting.** We use boldface uppercase letters to denote random vectors (e.g., $\mathbf{X}$) and lowercase for their values (e.g., $\mathbf{x}$). For a subset $S \subseteq [d] := \{1, \ldots, d\}$, we denote its cardinality by $|S|$ and its complement $S^c = [d] \setminus S$. Subscripts index features; e.g., $\mathbf{x}_S$ represents $\mathbf{x}$ restricted to the entries indexed by $S$. We consider a supervised learning setting with an unknown distribution $\mathcal{D}$ over features $\mathcal{X} \subseteq \mathbb{R}^d$ and labels $\mathcal{Y} \subseteq \mathbb{R}$. We assume access to a model $f : \mathcal{X} \mapsto \mathcal{Y}$ that was trained on samples from $\mathcal{D}$. For an input $\mathbf{x} = (x_1, \ldots, x_d) \in \mathbb{R}^d$, the goal is to identify the important features in $\mathbf{x}$ for the prediction $f(\mathbf{x})$. To define importance, we will use the average restricted prediction, $f_S(\mathbf{x}) = \mathbb{E}_{\mathbf{X}_{S^c} \sim \mathcal{V}_{S^c}} [f(\mathbf{x}_S, \mathbf{X}_{S^c})]$, where $\mathbf{x}_S$ is fixed and $\mathbf{X}_{S^c}$ is a random vector drawn from an arbitrary reference distribution $\mathcal{V}_{S^c}$ (which may or may not depend on $S^c$). For example, two common choices are the marginal $\mathcal{V}_{S^c} = p(\mathbf{X}_{S^c})$ and conditional distribution $\mathcal{V}_{S^c} = p(\mathbf{X}_{S^c} \mid \mathbf{x}_S)$. This strategy, popularized in (Lundberg & Lee, 2017; Lundberg et al., 2020), allows us to query $f$, which only takes inputs in $\mathbb{R}^d$, and analyze its behavior when sets of features are retained or removed.

**Definitions.** We now present our proposed definitions of sufficiency and necessity. At a high level, these definitions were formalized to align with the following guiding principles:

P1. $S$ is sufficient if it is enough to generate the original prediction, i.e. $f_S(\mathbf{x}) \approx f(\mathbf{x})$.

P2. $S$ is necessary if we cannot generate the original prediction without it, i.e. $f_{S^c}(\mathbf{x}) \not\approx f(\mathbf{x})$.

P3. The set $S = [d]$ should be maximally sufficient and necessary for $f(\mathbf{x})$.

The principles P1 and P2 are natural and agree with the logical notions of sufficiency and necessity. Furthermore, because the full set of features provides all the information needed to make the prediction $f(\mathbf{x})$, it should thus be regarded as maximally sufficient and necessary (P3). With these principles laid out, we now formally define sufficiency and necessity.

**Definition 2.1** (Sufficiency). *Let $\epsilon \geq 0$ and let $\rho : \mathbb{R} \times \mathbb{R} \mapsto \mathbb{R}$ be a metric on $\mathbb{R}$. A subset $S \subseteq [d]$ is $\epsilon$-sufficient with respect to a distribution $\mathcal{V}$ for $f$ at $\mathbf{x}$ if*

$$\Delta_{\mathcal{V}}^{suf}(S, f, \mathbf{x}) \triangleq \rho(f(\mathbf{x}), f_S(\mathbf{x})) \leq \epsilon. \tag{1}$$

*Furthermore, $S$ is $\epsilon$-super sufficient if all supersets $\widetilde{S} \supseteq S$ are $\epsilon$-sufficient.*

This notion of sufficiency is straightforward and aligns with P1. A subset $S$ is $\epsilon$-sufficient with respect to reference distribution $\mathcal{V}$ if, with $\mathbf{x}_S$ fixed, the average restricted prediction $f_S(\mathbf{x})$ is within $\epsilon$ from the original $f(\mathbf{x})$. Furthermore, $S$ is $\epsilon$-super sufficient if $\rho(f(\mathbf{x}), f_S(\mathbf{x})) \leq \epsilon$ and, $\forall \widetilde{S} \supseteq S$,

$\rho(f(\mathbf{x}), f_{\widetilde{S}}(\mathbf{x})) \leq \epsilon$. Namely, including more features in $S$ keeps $f_S(\mathbf{x})$ $\epsilon$ close to $f(\mathbf{x})$. Note this definition aligns with P3, since the set $S = [d]$ is 0-sufficient (maximally sufficient). To find a small sufficient subset $S$ of small cardinality $\tau > 0$, we can solve the following optimization problem:

$$\underset{S \subseteq [d]}{\arg\min} \ \Delta_{\mathcal{V}}^{\mathsf{suf}}(S, f, \mathbf{x}) \ \text{ subject to } \ |S| \leq \tau \tag{$P_{\mathsf{suf}}$}$$

We will refer to this problem as the *sufficiency problem*, or ($P_{\mathsf{suf}}$). Using analogous ideas, we also define necessity and formulate an optimization problem to find small necessary subsets.

**Definition 2.2** (Necessity). *Let $\epsilon \geq 0$ and denote $\rho : \mathbb{R} \times \mathbb{R} \mapsto \mathbb{R}$ to be metric on $\mathbb{R}$. A subset $S \subseteq [d]$ is $\epsilon$-necessary with respect to a distribution $\mathcal{V}$ for $f$ at $\mathbf{x}$ if*

$$\Delta_{\mathcal{V}}^{nec}(S, f, \mathbf{x}) \triangleq \rho(f_{S^c}(\mathbf{x}), f_{\emptyset}(\mathbf{x})) \leq \epsilon. \tag{2}$$

*Furthermore, $S$ is $\epsilon$-super necessary if all supersets $\widetilde{S} \supseteq S$ are $\epsilon$-necessary.*

Here, a subset $S$ is $\epsilon$-necessary if marginalizing out the features in $S$ with respect to $\mathcal{V}_S$, results in an average restricted prediction $f_{S^c}(\mathbf{x})$ that is $\epsilon$ close to $f_{\emptyset}(\mathbf{x})$ – the average baseline prediction of $f$ over $\mathcal{V}_{[d]}$. Furthermore, $S$ is $\epsilon$-super necessary if $\rho(f_S(\mathbf{x}), f(\mathbf{x})) \leq \epsilon$ and, $\forall \widetilde{S} \supseteq S$, $\epsilon$-necessary. Note, our definition of necessity differs from alternatives (Dhurandhar et al., 2018; Pawelczyk et al., 2020) which state that $S$ is necessary if $\rho(f(\mathbf{x}), f_{S^c}(\mathbf{x})) \geq \Delta$ for some $\Delta > 0$. Our notion is more general in that it implies this condition. Intuitively, if $f_{\emptyset}(\mathbf{x})$ and $f(\mathbf{x})$ differ, and $f_{S^c}(\mathbf{x})$ is close to $f_{\emptyset}(\mathbf{x})$, then $f_{S^c}(\mathbf{x})$ and $f(\mathbf{x})$ will also differ. Furthermore, for $S = [d]$, we have $\Delta^{nec}\mathcal{V}(S, f, \mathbf{x}) \triangleq \rho(f\emptyset(\mathbf{x}), f_{\emptyset}(\mathbf{x})) = 0$, indicating that $S = [d]$ is 0-necessary (maximally necessary) as desired. A detailed comparison of our approach with classical definitions, along with its advantages, is provided in the Appendix. To identify a $\epsilon$-necessary subset $S$ of small cardinality $\tau > 0$, one can solve the following optimization problem, which we refer to as the *necessity* problem or ($P_{\mathsf{nec}}$).

$$\underset{S \subseteq [d]}{\arg\min} \ \Delta_{\mathcal{V}}^{\mathsf{nec}}(S, f, \mathbf{x}) \ \text{ subject to } \ |S| \leq \tau \tag{$P_{\mathsf{nec}}$}$$

Having presented our definitions, we now discuss related works before presenting our main results.

## 3  RELATED WORK

Notions of sufficiency, necessity, their duality and connections with other feature attribution methods have been studied to varying degrees. We comment on the main related works in this section.

**Sufficiency.** The notion of sufficient features has gained significant attention in recent research. Shih et al. (2018) explore a symbolic approach to explain Bayesian network classifiers and introduce prime implicant explanations, which are minimal subsets $S$ that make features in the complement irrelevant to the prediction $f(\mathbf{x})$. For models represented by a finite set of first-order logic (FOL) sentences, Ignatiev et al. (2020) refer to prime implicants as abductive explanations (AXp's). For classifiers defined by propositional formulas and inputs with discrete features, Darwiche & Hirth (2020) refer to prime implicants as sufficient reasons and define a complete reason to be the disjunction of all sufficient reasons. They present efficient algorithms, leveraging Boolean circuits, to compute sufficient and complete reasons and demonstrate their use in identifying classifier dependence on protected features that should not inform decisions. For more complex models, Ribeiro et al. (2018) propose high-precision probabilistic explanations called anchors, which represent local, sufficient conditions. For $\mathbf{x}$ positively classified by $f$, Wang et al. (2021) propose a greedy approach to solve ($P_{\mathsf{suf}}$), I Amoukou & Brunel (2022) extend this work to regression settings using tree-based models, and Fong & Vedaldi (2017) introduce the preservation method which relaxes $S$ to $[0,1]^d$.

**Necessity.** There has also been significant focus on identifying necessary features – those that, when altered, lead to a change in the prediction $f(\mathbf{x})$. For models expressible by FOL sentences, Ignatiev et al. (2019) define prime implicates as the minimal subsets that when changed, modify the prediction $f(\mathbf{x})$ and relate these to adversarial examples. For Boolean models predicting on samples $\mathbf{x}$ with discrete features, Ignatiev et al. (2020) and (Darwiche & Hirth, 2020) refer to prime implicates as contrastive explanations (CXp's) and necessary reasons, respectively. Beyond boolean functions, for $\mathbf{x}$ positively classified by a classifier $f$, Fong et al. (2019) relax $S$ to $[0,1]^d$ and propose the deletion method to approximately solve ($P_{\mathsf{nec}}$).

**Duality Between Sufficiency and Necessity.** Dabkowski & Gal (2017) characterize the preservation and deletion methods as discovering the *smallest sufficient* and *destroying region* (SSR and SDR).

They propose combining the two but do not explore how solutions to this approach may differ from individual SSR and SDR solutions. Ignatiev et al. (2020) show that AXp's and CXp's are minimal hitting sets of another by using a hitting set duality result between minimal unsatisfiable and correction subsets. The result enables the identification of AXp's from CXp's and vice versa.

**Sufficiency, Necessity, and General Feature Attribution Methods.** Precise connections between sufficiency, necessity, and other popular feature attribution methods (such as Shapley values (Shapley, 1951; Chen et al., 2018b; Lundberg & Lee, 2017)) remains unclear. To our knowledge, Covert et al. (2021) provide the only work examining these approaches (Fong & Vedaldi, 2017; Fong et al., 2019; Dabkowski & Gal, 2017) in the context of general removal-based methods, i.e., methods that remove certain input features to evaluate different notions of importance. The work of Watson et al. (2021) is also relevant to our work, as it formalizes a connection between notions of sufficiency and Shapley values. With the specific payoff function defined as $v(S) = \mathbb{E}[f(\mathbf{x}_S, \mathbf{X}_{S^c})]$, they show how each summand in the Shapley value measures the sufficiency of feature $i$ to a particular subset.

## 4 UNIFYING SUFFICIENCY AND NECESSITY

Given a model $f$ and sample $\mathbf{x}$, we can identify a small set of important features $S$ by solving either (P$_{\sf suf}$) or (P$_{\sf nec}$). While both methods are popular (Kolek et al., 2022; Fong & Vedaldi, 2017; Bhalla et al., 2023; Yoon et al., 2018). identifying small sufficient or necessary subsets may not provide a complete picture of how $f$ uses $\mathbf{x}$ to make a prediction. To see why, consider the following scenario: for a fixed $\tau > 0$, let $S^*$ be a $\epsilon$-sufficient solution to (P$_{\sf suf}$), so that $|S^*| \leq \tau$ and $\Delta_{\mathcal{V}}^{\sf suf}(S, f, \mathbf{x}) \leq \epsilon$. While $S^*$ is $\epsilon$-sufficient, it can also be true that $\Delta_{\mathcal{V}}^{\sf nec}(S, f, \mathbf{x}) > \epsilon$ indicating $S^*$ is **not** $\epsilon$-necessary: indeed, this can simply happen when its complement, $S^{c*}$, contains important features. This scenario raises two questions: 1) How different are sufficient and necessary features? 2) How does varying the levels of sufficiency and necessity affect the optimal set of important features?

To answer these important questions (and avoid the scenario above) we propose studying a unification of (P$_{\sf suf}$) and (P$_{\sf nec}$).Consider $\Delta_{\mathcal{V}}^{\sf uni}(S, f, \mathbf{x}, \alpha) = \alpha \cdot \Delta_{\mathcal{V}}^{\sf suf}(S, f, \mathbf{x}) + (1 - \alpha) \cdot \Delta_{\mathcal{V}}^{\sf nec}(S, f, \mathbf{x})$, a convex combination of $\Delta_{\mathcal{V}}^{\sf suf}(S, f, \mathbf{x})$ and $\Delta_{\mathcal{V}}^{\sf nec}(S, f, \mathbf{x})$, where $\alpha \in [0, 1]$ controls the extent to which $S$ is sufficient vs. necessary. Our *unified problem*, (P$_{\sf uni}$), can be expressed as:

$$\underset{S \subseteq [d]}{\arg\min} \ \Delta_{\mathcal{V}}^{\sf uni}(S, f, \mathbf{x}, \alpha) \ \text{ subject to } \ |S| \leq \tau \qquad \text{(P}_{\sf uni}\text{)}$$

When $\alpha$ is 1 or 0, $\Delta_{\mathcal{V}}^{\sf uni}(S, f, \mathbf{x}, \alpha)$ reduces to $\Delta_{\mathcal{V}}^{\sf suf}(S, f, \mathbf{x})$ or $\Delta_{\mathcal{V}}^{\sf nec}(S, f, \mathbf{x})$, respectively. In these extreme cases, $S$ is only sufficient or necessary. In the remainder of this work we will theoretically analyze (P$_{\sf uni}$), characterize its solutions, and provide different interpretations of what properties the solutions have through the lens of conditional independence and game theory. In the experimental section, we will show that solutions to (P$_{\sf uni}$) provide insights that neither (P$_{\sf suf}$) nor (P$_{\sf nec}$) offer.

### 4.1 SOLUTIONS TO THE UNIFIED PROBLEM

We begin with a simple lemma that demonstrates why (P$_{\sf uni}$) enforces both sufficiency and necessity.

**Lemma 4.1.** *Let $\alpha \in (0, 1)$. For $\tau > 0$, denote $S^*$ to be a solution to* (P$_{\sf uni}$) *for which $\Delta_{\mathcal{V}}^{uni}(S, f, \mathbf{x}, \alpha) = \epsilon$. Then, $S^*$ is $\frac{\epsilon}{\alpha}$-sufficient and $\frac{\epsilon}{1-\alpha}$-necessary. Formally,*

$$0 \leq \Delta_{\mathcal{V}}^{suf}(S^*, f, \mathbf{x}) \leq \frac{\epsilon}{\alpha} \quad and \quad 0 \leq \Delta_{\mathcal{V}}^{nec}(S^*, f, \mathbf{x}) \leq \frac{\epsilon}{1-\alpha}. \qquad (3)$$

The proof of this result, and all others, is included Appendix A.1. This result illustrates that solutions to (P$_{\sf uni}$) satisfy varying definitions of sufficiency and necessity. Furthermore, as $\alpha$ increases from 0 to 1, the solution shifts from being highly necessary to highly sufficient. In the following results, we will show *when* and *how* solutions to (P$_{\sf uni}$) are similar (and different) to those of (P$_{\sf suf}$) and (P$_{\sf nec}$). To start, we present the following lemma, which will be useful in subsequent results.

**Lemma 4.2.** *For $0 \leq \epsilon < \frac{\rho(f(\mathbf{x}), f_\emptyset(\mathbf{x}))}{2}$, denote $S_{suf}^*$ and $S_{nec}^*$ to be $\epsilon$-sufficient and $\epsilon$-necessary sets. Then, if $S_{suf}^*$ is $\epsilon$-super sufficient or $S_{nec}^*$ is $\epsilon$-super necessary, we have $S_{suf}^* \cap S_{nec}^* \neq \emptyset$.*

This lemma demonstrates that, given $\epsilon$-sufficient and necessary sets $S_{\sf suf}^*$ and $S_{\sf nec}^*$, if either additionally satisfies the stronger notions of super sufficiency or necessity, they must share some features. This proves useful in characterizing a solution to (P$_{\sf uni}$), which we now do in the following theorem.

**Theorem 4.1.** *Let $\tau_1, \tau_2 > 0$ and $0 \le \epsilon < \frac{1}{2} \cdot \rho(f(\mathbf{x}), f_\emptyset(\mathbf{x}))$. Denote $S^*_{suf}$ and $S^*_{nec}$ to be $\epsilon$-super sufficient and $\epsilon$-super necessary solutions to $(P_{suf})$ and $(P_{nec})$, respectively, such that $|S^*_{suf}| = \tau_1$ and $|S^*_{nec}| = \tau_2$. Then, there exists a set $S^*$ such that*

$$\Delta^{uni}_{\mathcal{V}}(S^*, f, \mathbf{x}, \alpha) \le \epsilon \quad and \quad \max(\tau_1, \tau_2) \le |S^*| < \tau_1 + \tau_2. \tag{4}$$

*Furthermore, if $S^*_{suf} \subseteq S^*_{nec}$ or $S^*_{nec} \subseteq S^*_{suf}$, then $S^* = S^*_{nec}$ or $S^* = S^*_{suf}$, respectively.*

This result demonstrates that when there are $\epsilon$-super sufficient and $\epsilon$-super necessary solutions to $(P_{suf})$ and $(P_{nec})$, then one can identify a set $S^*$ with small $\Delta^{uni}$. As an example, consider features that are $\epsilon$-super sufficient, $S^*_{suf}$. If we have domain knowledge that $S^*_{suf} \subseteq S^*_{nec}$, and $S^*_{nec}$ is $\epsilon$-super necessary, then $S^*_{nec}$ will have a small $\Delta^{uni}$ Conversely, if we know that $S^*_{suf}$ is $\epsilon$-super necessary along with being a subset of $\epsilon$-super sufficient set $S^*_{suf}$, then $S^*_{suf}$ will have a small $\Delta^{uni}$.

## 5  Two Perspectives of the Unified Approach

In the previous section, we characterized solutions to $(P_{uni})$ and their connections to those of $(P_{suf})$ and $(P_{nec})$. To further motivate and the unified approach, we now offer two alternative perspectives of our framework through the lens of conditional independence and Shapley values.

### 5.1  A Conditional Independence Perspective

Here we demonstrate how sufficiency, necessity, and their unification, can be understood as conditional independence relations between features $\mathbf{X}$ and label $Y$.

**Corollary 5.1.** *Suppose $\forall S \subseteq [d]$, $\mathcal{V}_S = p(\mathbf{X}_S | \mathbf{X}_{S^c} = \mathbf{x}_{S^c})$. Let $\alpha \in (0, 1)$, $\epsilon \ge 0$, and denote $\rho : \mathbb{R} \times \mathbb{R} \mapsto \mathbb{R}$ to be a metric. Furthermore, for $\tau > 0$ and $f(\mathbf{X}) = \mathbb{E}[Y \mid \mathbf{X}]$, let $S^*$ be a solution to $(P_{uni})$ such that $\Delta^{uni}_{\mathcal{V}}(S, f, \mathbf{x}, \alpha) = \epsilon$. Then, $S^*$ satisfies the follow conditional independencies,*

$$\rho\left(\mathbb{E}[Y \mid \mathbf{x}], \mathbb{E}[Y \mid \mathbf{X}_{S^*} = \mathbf{x}_{S^*}]\right) \le \frac{\epsilon}{\alpha} \quad and \quad \rho\left(\mathbb{E}[Y \mid \mathbf{X}_{S^*_c} = \mathbf{x}_{S^*_c}], \mathbb{E}[Y]\right) \le \frac{\epsilon}{1 - \alpha}. \tag{5}$$

The assumption in this corollary is that, $\forall S \subseteq [d]$, $f_S(\mathbf{x})$ is evaluated using the conditional distribution $p(\mathbf{X}_{S^c} \mid \mathbf{X}_S = \mathbf{x}_S)$ as the reference distribution $\mathcal{V}_S$. Given the recent advancements in generative models (Song & Ermon, 2019; Ho et al., 2020; Song et al., 2021), this assumption is (approximately) reasonable in many practical settings, as we will demonstrate in our experiments. For this particular $\mathcal{V}_S$, the result shows that minimizing $(P_{uni})$ with model $f(\mathbf{X}) = \mathbb{E}[Y \mid \mathbf{X}]$ identifies an $S^*$ that approximately satisfies two conditional independence properties. First, $S^*$ is sufficient as conditioning on $S^*$ leaves the complement $S^{c*}$ with minimal additional information about $Y$. Second, $S^*$ is necessary because when we solely rely on the complement $S^{c*}$, the information gained about $Y$ is minimal and similar to $\mathbb{E}[Y = 1]$.

### 5.2  A Shapley Value Perspective

In the previous section, we detailed the conditional independence relations being optimized for when solving $(P_{uni})$. We now present an arguably less intuitive result that shows that solving $(P_{uni})$ is equivalent to maximizing the lower bound of the Shapley value. Before presenting our result, we provide a brief background on this game-theoretic quantity.

**Shapley Values.** Shapley values use game theory to measure the importance of players in a game. Let the tuple $([n], v)$ represent a cooperative game with players $[n] = \{1, 2, \ldots, n\}$ and denote a characteristic function $v(S) : \mathcal{P}([n]) \to \mathbb{R}$, Then, the Shapley value (Shapley, 1951) for player $j$ in the game $([n], v)$ is $\phi^{\text{shap}}_j([n], v) = \sum_{S \subseteq [n] \setminus \{j\}} w_S \cdot [v(S \cup \{j\}) - v(S)]$ where $w_S = \frac{|S|!(n - |S| - 1)!}{n!}$. In the context of XAI, Shapley values are widely used to measure local feature importance by treating input features as players in a game (Covert et al., 2020; Teneggi et al., 2022; Chen et al., 2018b; Lundberg & Lee, 2017). Given a sample $\mathbf{x}$ and a model $f$, the importance of $x_j$ to the prediction $f(\mathbf{x})$ is measured by computing $\phi^{\text{shap}}_j$ for a game $([d], v)$, where $v(S)$ quantifies how the features in $S$ contribute to $f(\mathbf{x})$. Different choices of $v(S)$ can be found in (Lundberg & Lee, 2017; Sundararajan & Najmi, 2020; Watson et al., 2024). Although computing $\phi^{\text{shap}}_j$ is computationally intractable, several practical methods for estimation have been developed (Chen et al., 2023; Teneggi et al., 2022; Zhang et al., 2023; Lundberg et al., 2020). While Shapley values are popular across various domains (Moncada-Torres et al., 2021; Zoabi et al., 2021; Liu et al., 2021), few works, aside from Watson et al. (2021), explore their connections to sufficiency and necessity.

With this background, we now present our result. Recall solving (P$_{uni}$) finds a small subset $S$ with low $\Delta_{\mathcal{V}}^{uni}(S, f, \mathbf{x}, \alpha)$. Notice that (P$_{uni}$) naturally *partitions* the features into two sets, $S$ and $S^c$. In the following theorem we demonstrate that finding a small $S$ with minimal $\Delta_{\mathcal{V}}^{uni}(S, f, \mathbf{x}, \alpha)$ is equivalent to maximizing a lower bound on the Shapley value in a two player game.

**Theorem 5.1.** *Consider an input $\mathbf{x}$ for which $f(\mathbf{x}) \neq f_\emptyset(\mathbf{x})$. Denote by $\Lambda_d = \{S, S^c\}$ the partition of $[d] = \{1, 2, \ldots, d\}$, and define the characteristic function to be $v(S) = -\rho(f(\mathbf{x}), f_S(\mathbf{x}))$. Then,*

$$\phi_S^{shap}(\Lambda_d, v) \geq \rho(f(\mathbf{x}), f_\emptyset(\mathbf{x})) - \Delta_{\mathcal{V}}^{uni}(S, f, \mathbf{x}, \alpha). \tag{6}$$

This result motivates minimizing $\Delta_{\mathcal{V}}^{uni}(S, f, \mathbf{x}, \alpha)$ via a game-theoretic interpretation. The tuple $(\Lambda_d, v)$ specifies a game, and since there are $2^{d-1}$ ways to partition $[d]$ into 2 subsets, there are $2^{d-1}$ games. The inequality above holds for each of them. Thus, Theorem 5.1 implies that finding the $S$ with minimal $\Delta_{\mathcal{V}}^{uni}(S, f, \mathbf{x}, \alpha)$ is equivalent to identifying the game (i.e. partition) $(\Lambda_d, v)$ in which $S$ has the largest lower bound on its Shapley value.

# 6 SOLVING THE UNIFIED PROBLEM

Before presenting our results, we briefly discuss different approaches to solving (P$_{uni}$). In general, this problem is NP-hard however, in certain settings, one can efficiently compute exact solutions or use tractable relaxations, (Kolek et al., 2022; Fong et al., 2019; Linder et al., 2022) to approximate solutions. We present these general approaches here, and defer details to Appendix A.2.

**Exhaustive Search.** When the feature space dimension, $d$, or choice of $\tau \in \mathbb{Z}_{>0}$ is small an exhaustive search can compute exact solutions to (P$_{uni}$) by evaluating $\Delta_{\mathcal{V}}^{uni}(S, f, \mathbf{x}, \alpha)$ for all $\binom{d}{\tau}$ subsets $S$ of cardinality $\tau$ and selecting the minimizer.

**Instance-wise Optimization.** When $d$ is large, rendering (P$_{uni}$) intractable, one can generate approximate solutions by solving the relaxed problem[1]

$$\arg \min_{S \subseteq [0,1]^d} \Delta_{\mathcal{V}}^{uni}(S, f, \mathbf{x}, \alpha) + \lambda_1 \cdot ||S||_1 + \lambda_{TV} \cdot ||S||_{TV}. \tag{7}$$

This type of approach is often used in computer vision and natural language problems (Fong et al., 2019; Kolek et al., 2022; Linder et al., 2022) to generate instance-specific solutions.

**Parametric Model Approach.** Another we approach we take to generate solutions to (P$_{uni}$) is to learn models $g_\theta : \mathcal{X} \mapsto [0, 1]^d$ that (approximately) solve the following optimization problem:

$$\arg \min_{\theta \in \Theta} \mathbb{E}_{\mathbf{X} \sim \mathcal{D}_\mathcal{X}} \left[ \Delta_{\mathcal{V}}^{uni}(g_\theta(\mathbf{X}), f, \mathbf{X}, \alpha) + \lambda_1 \cdot ||g_\theta(\mathbf{X})||_1 + \lambda_{TV} \cdot ||g_\theta(\mathbf{X})||_{TV} \right]. \tag{8}$$

With these models, an approximate solution can be computed via $g_\theta(\mathbf{x})$. This method is popular (Chen et al., 2018a; Yoon et al., 2018; Linder et al., 2022), as it handles highly structured data well and requires training only one model, rather than repeatedly solving Eq. (7) for each sample.

# 7 EXPERIMENTS

We demonstrate our theoretical findings in multiple settings of increasingly complexity: two tabular data tasks (on synthetic data and the US adult income dataset (Ding et al., 2021)) and two high-dimensional image classification tasks using the RSNA 2019 Brain CT Hemorrhage Challenge (Flanders et al., 2020) and CelebA-HQ datasets (Lee et al., 2020)

## 7.1 TABULAR DATA

With the following tabular data settings, we demonstrate how the specific trade-off between sufficiently and necessity can greatly alter the solutions to (P$_{uni}$). To do so, we compute exact solutions via exhaustive search to (P$_{uni}$) for varying levels of sufficiency vs. necessity and multiple size constraints. We learn a predictor $f$ and, for 100 new samples, solve (P$_{uni}$) for $\tau \in \{3, 6, 9\}$ and $\alpha \in [0, 1]$, with $\rho(a, b) = |a - b|$ and $\mathcal{V}_S = p(\mathbf{X}_S \mid \mathbf{X}_{S^c} = \mathbf{x}_{S^c})$. For a fixed $\tau$ and sample $\mathbf{x}$, we denote $S_{\alpha_i}^*$ to be a solution to (P$_{uni}$) for $\alpha_i$. It is represented as a binary vector $s \in \{0, 1\}^{10}$, where $s_j = 1$ if $j \in S_{\alpha_i}^*$ and 0 otherwise. To analyze the stability of $S_{\alpha_i}^*$ as sufficiency and necessity vary, we report the normalized average Hamming distance (Hamming, 1950) between $S_{\alpha_i}^*$ and $S_0^*$ (with 95% confidence intervals) as a function of $\alpha$.

---

[1]Here, $\lambda_1$, $||S||_1$ and $\lambda_{TV}$, $||S||_{TV}$ are the $\ell_1$ and Total Variation norms and hyperparamters, respectively, promoting sparsity and smoothness.

### 7.1.1 LINEAR REGRESSION

We begin with a regression example. Features are distributed as $\mathbf{X} \sim \mathcal{N}(\boldsymbol{\mu}, \mathbf{A}\mathbf{A}^{\mathbf{T}})$ with $\boldsymbol{\mu} = \left[2^i\right]_{i=1}^d$ and $\mathbf{A}_{i,j} \sim U(0,1)$. The response is $Y = \boldsymbol{\beta}^T\mathbf{X} + \boldsymbol{\epsilon}$, with $\boldsymbol{\beta} = 32 \cdot [2^{-i}]_{i=1}^d$ and $\boldsymbol{\epsilon} \sim \mathcal{N}(\mathbf{0}, \mathbf{I}_{d \times d})$. We fix $d = 10$ and use the model $f(\mathbf{X}) = \hat{\boldsymbol{\beta}}^T\mathbf{X}$, where $\hat{\boldsymbol{\beta}}$ is the least squares solution.

**Stability of Unified Solutions.** Fig. 1a shows that when solutions are constrained to be small ($\tau = 3$), increasing $\alpha$ to enforce greater sufficiency results in a steady increase in Hamming distance, indicating that the solutions $S_{\alpha_i}^*$ are consistently changing. When larger solutions are allowed ($\tau = 6$), $S_{\alpha_i}^*$ rapidly changes with the introduction of sufficiency, as seen by the initial steep rise in Hamming distance. However, as $\alpha$ continues to increase, this distance grows more gradually. Lastly, when the solution size approaches the dimension of the feature space ($\tau = 9$), small to medium levels of sufficiency do not significantly alter $S_{\alpha_i}^*$. However, high levels of sufficiency ($\alpha > 0.8$) lead to extreme changes in the solutions, as shown by a sharp increase in Hamming distance.

### 7.1.2 AMERICAN COMMUNITY SURVEY INCOME (ACSINCOME)

We use the ACSIncome dataset for California, including 10 demographic and socioeconomic features such as age, education, occupation, and geographic region. We train a Random Forest classifier to predict whether an individual's annual income exceeds \$50K, achieving a test accuracy $\approx 81\%$.

**Stability of Unified Solutions.** Fig. 1b shows that when solutions are forced to be small ($\tau = 3$), increasing $\alpha$ to enforce sufficiency results in a steady increase in Hamming distance, indicating the solutions $S_{\alpha_i}^*$ are changing. For larger solutions ($\tau = 6$), $S_{\alpha_i}^*$ changes significantly when low levels sufficiency are required, indicated by initial rise in the Hamming distance. As $\alpha$ continues to increase, the Hamming distance grows more gradually. Interestingly, when the size is close to feature space's dimensionality ($\tau = 9$), the Hamming distance exhibits a behavior similar to that observed for $\tau = 3$. In conclusion, both examples show that the optimal feature set can vary depending on the size constraint and balance between sufficiency and necessity.

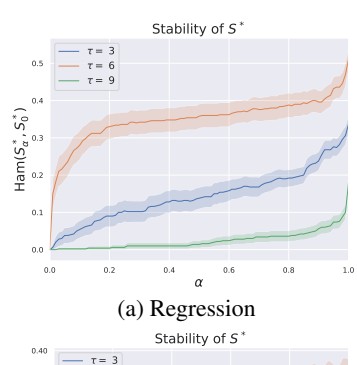

(a) Regression

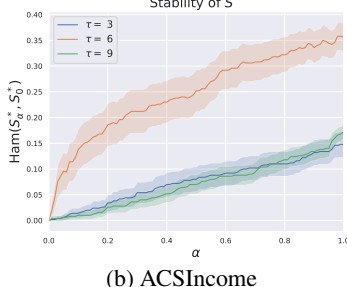

(b) ACSIncome

Figure 1: Stability of ($P_{\mathsf{uni}}$) Solutions

### 7.2 IMAGE CLASSIFICATION

The following two experiments explore high dimensional image classification tasks. The features are pixel values and so a subset $S$ corresponds to a binary mask identifying important pixels. Since solving ($P_{\mathsf{suf}}$), ($P_{\mathsf{nec}}$), or ($P_{\mathsf{uni}}$) is intractable here, we use two methods, the per-sample and model based approach in Eqs. (7) and (8) to identify sufficient and necessary masks. These experiments serve two purposes. First, they will analyze the ability popular explanation methods–including Integrated Gradients (Sundararajan et al., 2017), GradientSHAP (Lundberg & Lee, 2017), Guided GradCAM (Selvaraju et al., 2017), and h-Shap (Teneggi et al., 2022)–to identify small sufficient and necessary subsets. To ensure consistent analysis, all attribution scores are normalized to the interval $[0, 1]$. This is done by setting the top 1% of nonzero scores to 1 and dividing the remaining by the minimum score from the top 1% nonzero scores, which is common practice (Kokhlikyan et al., 2020). Binary masks are then generated by thresholding the normalized scores using thresholds $t \in (0, 1)$. For a test set of images and normalized attribution scores, we report the average (across all binary masks) $-\log(\Delta^{\mathsf{suf}})$, $-\log(\Delta^{\mathsf{nec}})$, and $-\log(L^0)$ where $L^0$ is the relative size of $S$ for $t \in (0, 1)$ to analyze the sufficiency, necessity and size of the explanations. The second objective of these experiments is to understand and visualize the similarities and differences between sufficient and necessary sets.

### 7.2.1 RSNA CT HEMORRHAGE

We use the RSNA 2019 Brain CT Hemorrhage Challenge dataset comprised of 752,803 scans. Each scan is annotated by expert neuroradiologists with the presence and type(s) of hemorrhage (i.e., epidural, intraparenchymal, intraventricular, subarachnoid, or subdural). We use a ResNet18 (He et al., 2016) classifier that was pretrained on this data (Teneggi et al., 2022). Since the dataset

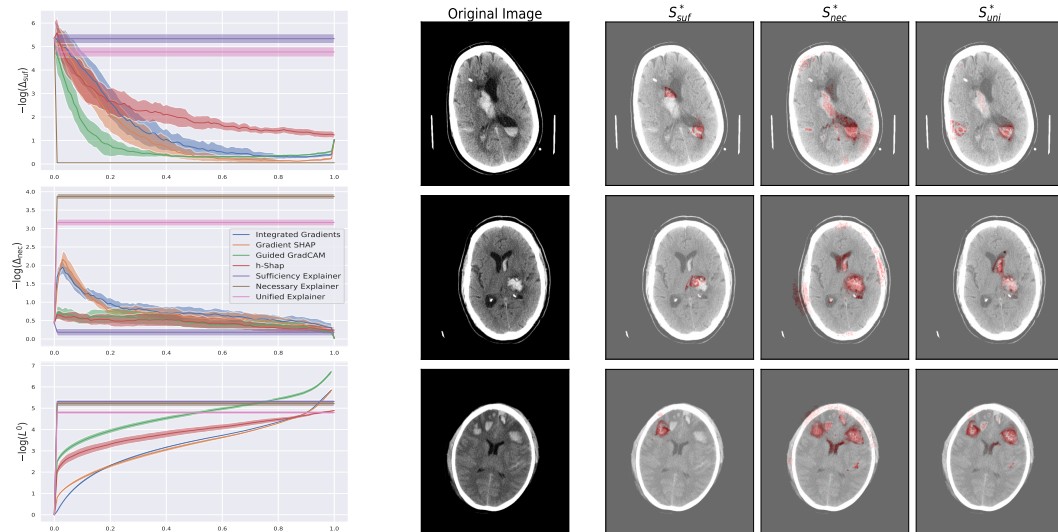

(a) Comparison of different methods.

(b) $S_{\mathsf{suf}}^*$, $S_{\mathsf{nec}}^*$ and $S_{\mathsf{uni}}^*$ for various CT scans.

Figure 2: Experimental results on the RSNA dataset.

consists of highly complex and diverse images, we employ the per-example approach in Eq. (7) with $\alpha \in \{0, 0.5, 1\}$ to learn sufficient and necessary masks. Further details are in Appendix A.2.

**Comparison of Post-hoc Interpretability Methods.** For a set of 20 images positively classified by the ResNet model, we apply multiple post-hoc interpretability methods, as well as compute sufficient and necessary masks by solving (7). The results in Fig. 2a show that for thresholds $t < 0.1$, many methods identify sufficient sets smaller in size than the sufficient and unified explainer, as indicated by their large values of $-\log(\Delta^{\mathsf{suf}})$ and smaller values of $-\log(L^0)$. However, for $t > 0.1$, only the sufficient and unified explainer identify sufficient sets of a constant small size. Importantly, *no methods, besides the necessity and unified explainers, identify necessary sets*. Furthermore, as expected, the sufficient explainer does not identify necessary sets and vice versa. The unified explainer, as expected, identifies a sufficient and necessary set (at the cost of a larger set). In conclusion, while off-the-shelf methods can identify sufficient, they do not identify necessary sets for small thresholds.

**Sufficiency vs. Necessity.** In Fig. 2b we visualize the sufficient and necessary features in various CT scans. The first observation is that sufficient subsets do not provide a complete picture of which features are important. Notice for all the CT scans, a sufficient set, $S_{\mathsf{suf}}^*$ highlights one or two, but never all, brain hemorrhages in the scans. For example, in the last row, $S_{\mathsf{suf}}^*$ only contains the right frontal lobe parenchymal hemorrhages, which happens to be one of the larger hemorrhages present. On the other hand, necessary sets, $S_{\mathsf{nec}}^*$, contain parts of, sometimes entirely, *all* hemorrhages in the scans. In the last row, $S_{\mathsf{nec}}^*$ contains all multifocal parenchymal hemorrhages in both right and left frontal lobes, because when all these regions are masked, the model yields a prediction $\approx 0.64$– the prediction of the model on the mean image. Finally, notice in the 2nd and 3rd columns that $S_{\mathsf{nec}}^*$ and $S_{\mathsf{uni}}^*$ are nearly identical, which precisely demonstrate Lemma 4.1 and Theorem 4.1 in practice. First, since $S_{\mathsf{suf}}^*$ is super sufficient, $S_{\mathsf{suf}}^*$ and $S_{\mathsf{nec}}^*$, share common features. Second, visually $S_{\mathsf{suf}}^* \subseteq S_{\mathsf{nec}}^*$ holds approximately and so $S_{\mathsf{nec}}^* = S_{\mathsf{uni}}^*$. Through this experiment we are able to highlight the differences between sufficient and necessary sets, show how each contain important and complementary information, and demonstrate our theory holding in real world settings.

### 7.2.2 CELEBA-HQ

We use a modified version of the CelebA-HQ dataset (Karras, 2017) that contains 30,000 celebrity faces resized to 256×256 pixels. We train a ResNet18 to classify whether a celebrity is smiling, achieving a test accuracy $\approx 94\%$ and use the model based approach via solving Eq. (8) to generate sufficient and necessary masks. Given the structured nature of the dataset and the similarity of features across images, we use the model approach because it prevents overfitting to spurious signals

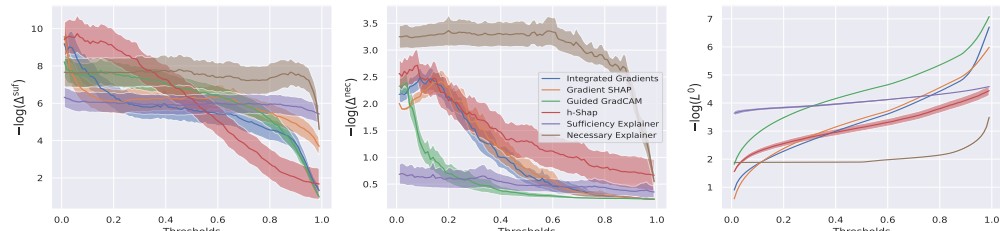

Figure 3: Comparison of different methods on the CelebAHQ dataset.

(Linder et al., 2022), an issue that can arise with per-example methods. Implementation details and hyperparameter settings are included in Appendix A.2.

**Comparison of Post-hoc Interpretability Methods.** For a set of 100 images labeled with a smile and correctly classified by the ResNet classifier, we apply multiple post-hoc interpretability methods and our sufficient and necessary explainers to identify important features associated with smiling. The results in Fig. 3 illustrate that for a wide range of thresholds $t \in [0, 1]$, many methods identify sufficient subsets, as $-\log(\Delta^{\mathsf{suf}})$ for many of them is comparable to that of the sufficient explainer. The necessary explainer, in fact, identifies subsets that are more sufficient than those found by the sufficient explainer. The reason is that the sufficient explainer identifies subsets that are, on average, smaller for all $t \in [0, 1]$, while the necessary explainer finds subsets that are constant in size for all $t \in [0, 1]$ but slightly larger since, to be necessary, they must contain more features that provide additional information about the label. For other methods, as $t$ increases, subset size decreases, and the sufficiency and necessity of the solutions decline. Meanwhile, the necessary explainer naturally identifies necessary subsets, indicated by large $-\log(\Delta^{\mathsf{nec}})$, whereas other methods fail to do so. In conclusion, many methods can identify sufficient sets, but not necessary ones and directly optimizing for these criterion leads to identifying small, constant-sized subsets across thresholds.

**Sufficiency vs. Necessity.** In Fig. 4, we see how sufficient subsets alone may overlook important features, while solutions to ($P_{\mathsf{uni}}$) offer deeper insights. As stated earlier, the sufficient explainer identifies sets that are sufficient but not necessary. On the other hand, the necessary explainer has high $-\log(\Delta^{\mathsf{suf}})$ and $-\log(\Delta^{\mathsf{nec}})$, indicating that it identifies sufficient *and* necessary set, meaning they also serve as solutions to ($P_{\mathsf{uni}}$). In Fig. 4, we visualize the reasons for this phenomena. Notice that $S^*_{\mathsf{suf}}$ precisely highlights (only) the smile. When $S^*_{\mathsf{suf}}$ is fixed, one can generate new images (as done in (Zhang et al., 2023)) for which the model produces the same predictions as it did for the original image (a smile). On the other hand, we also see why $S^*_{\mathsf{suf}}$ is *not* necessary: we can fix the complement $(S^*_{\mathsf{suf}})_c$ and, since there are important features in it, a smile is consistently generated, and the model produces the same prediction on these images as it did on the original. Conversely solutions to ($P_{\mathsf{nec}}$) (also solutions to ($P_{\mathsf{uni}}$) here) generate different explanations that provide a more complete picture of feature importance. Notice that $S^*_{\mathsf{nec}}$ is sufficient because $S^*_{\mathsf{suf}} \subseteq S^*_{\mathsf{nec}}$, with the additional features mainly being the dimples and eyes, which aid in determining the presence of a smile. More importantly, Fig. 5 illustrates why $S^*_{\mathsf{nec}}$ is necessary: when we fix the complement of $S^*_{\mathsf{nec}}$ and generate new samples, half of the faces lack a smile, leading the model $f$ to predict no smile. Additional images and details on sample generation are in Appendices A.2 and A.4.

## 8  LIMITATIONS & BROADER IMPACTS

While this work provides a novel theoretical contribution to the XAI community, there are some limitations that require careful discussion. The choice of reference distribution $\mathcal{V}_S$ determines the characteristics of sufficient and necessary explanations. For instance, only with the true conditional data distribution can one obtain the conditional independence results that our theory provides. Naturally, there are computational trade-offs that must be carefully studied; the ability to learn and sample from accurate conditional distributions to generate explanations with clear statistical meaning comes with a computational and statistical cost, particularly in high-dimensional settings. Thus, a key direction for future work is to explore the impact of different reference distributions and provide a principled framework for selecting a $\mathcal{V}_S$ that balances practical utility and computational feasibility.

Another relevant question is how well our proposed notions align with human intuition. While we aim to understand which features are sufficient and necessary *for a given predicted model*, these explanations may not always correspond to how humans perceive importance (since model might

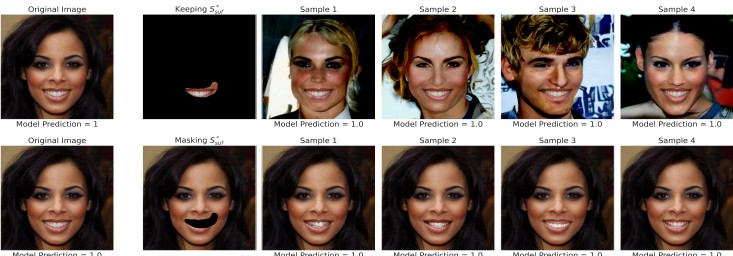

Figure 4: Images and model predictions by fixing and masking the sufficient subset $S_{\mathsf{suf}}^*$

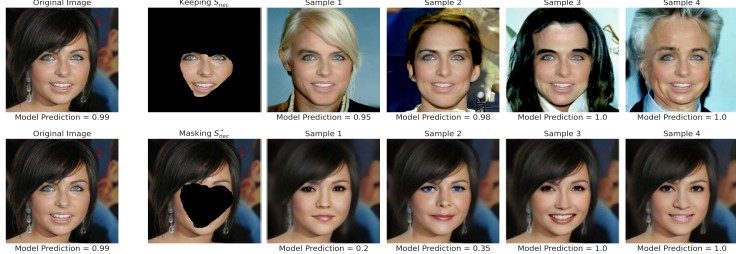

Figure 5: Images and model predictions by fixing and masking the necessary subset $S_{\mathsf{nec}}^*$

use different features to solve a task). This can be an issue in settings where interpretability is essential for trust and accountability, such as in healthcare. On the one hand, our approach can provide useful insights to further evaluate models (e.g. by verifying if the sufficient and necessary features employed by models correlate with the correct ones as informed by human experts). On the other hand, bridging the gap between our mathematical definitions of sufficiency and necessity and other human notions of importance is an area for further investigation. User studies, along with collaboration with domain experts, will be critical in determining how our formal notions of sufficiency and necessity can be adapted or extended to better meet real-world interpretability needs.

Finally, the societal impact of this work warrants discussion. While we offer a rigorous framework to understand model predictions, these are oblivious to notions of demographic bias (Hardt et al., 2016; Feldman et al., 2015; Bharti et al., 2024). There is a risk that an "incorrect" choice of generating a sufficient vs. necessary explanation could reinforce biases or obscure the causal reasons behind predictions. Future work will study when and how our framework can incorporate these biases.

## 9 CONCLUSION

This work formalizes notions of sufficiency and necessity as tools to evaluate feature importance and explain model predictions. We demonstrate that sufficient and necessary explanations, while insightful, often provide incomplete while complementary answers to model behavior. To address this limitation, we propose a unified approach that offers a new and more nuanced understanding of model behavior. Our unified approach expands the scope of explanations and reveals trade-offs between sufficiency and necessity, giving rise to new interpretations of feature importance. Through our theoretical contributions, we present conditions under which sufficiency and necessity align or diverge, and provide two perspectives of our unified approach through the lens of conditional independence and Shapley values. Our experimental results support our theoretical findings, providing examples of how adjusting sufficiency-necessity trade-off via our unified approach can uncover alternative sets of important features that would be missed by focusing solely on sufficiency or necessity. Furthermore, we evaluate common post-hoc interpretability methods showing that many fail to reliably identify features that are necessary or sufficient. In summary, our work contributes to a more complete understanding of feature importance through sufficiency and necessity. We believe, and hope, our framework holds potential for advancing the rigorous interpretability of ML models.

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

## A  APPENDIX

### A.1  PROOFS

#### A.1.1  PROOF OF LEMMA 4.1

**Lemma 4.1.** Let $\alpha \in (0,1)$. For $\tau > 0$, denote $S^*$ to be a solution to (P$_{\text{uni}}$) for which $\Delta_{\mathcal{V}}^{\text{uni}}(S^*, f, \mathbf{x}, \alpha) = \epsilon$. Then, $S^*$ is $\frac{\epsilon}{\alpha}$-sufficient and $\frac{\epsilon}{1-\alpha}$-necessary. Formally,

$$0 \leq \Delta_{\mathcal{V}}^{\text{suf}}(S^*, f, \mathbf{x}) \leq \frac{\epsilon}{\alpha} \quad \text{and} \quad 0 \leq \Delta_{\mathcal{V}}^{\text{nec}}(S^*, f, \mathbf{x}) \leq \frac{\epsilon}{1-\alpha}. \tag{9}$$

*Proof.* Let $\tau > 0$ and $\alpha \in (0,1)$ and denote $S^*$ to be a solution to (P$_{\text{uni}}$) such that

$$\Delta_{\mathcal{V}}^{\text{uni}}(S^*, f, \mathbf{x}, \alpha) = \epsilon. \tag{10}$$

Then, by definition of being a solution to (P$_{\text{uni}}$),

$$|S^*| \leq \tau. \tag{11}$$

Furthermore, recall that

$$\Delta_{\mathcal{V}}^{\text{uni}}(S^*, f, \mathbf{x}, \alpha) = \alpha \cdot \Delta_{\mathcal{V}}^{\text{suf}}(S^*, f, \mathbf{x}) + (1-\alpha) \cdot \Delta_{\mathcal{V}}^{\text{nec}}(S^*, f, \mathbf{x}) \tag{12}$$

which implies

$$\alpha \cdot \Delta_{\mathcal{V}}^{\text{suf}}(S^*, f, \mathbf{x}) = \epsilon - (1-\alpha) \cdot \Delta_{\mathcal{V}}^{\text{nec}}(S^*, f, \mathbf{x}) \tag{13}$$

$$\leq \epsilon \qquad\qquad ((1-\alpha),\ \Delta_{\mathcal{V}}^{\text{nec}}(S^*, f, \mathbf{x}) \geq 0) \tag{14}$$

$$\implies \Delta_{\mathcal{V}}^{\text{suf}}(S^*, f, \mathbf{x}) \leq \frac{\epsilon}{\alpha}. \tag{15}$$

Similarly,

$$(1-\alpha) \cdot \Delta_{\mathcal{V}}^{\text{nec}}(S^*, f, \mathbf{x}) = \epsilon - \alpha \cdot \Delta_{\mathcal{V}}^{\text{suf}}(S^*, f, \mathbf{x}) \tag{16}$$

$$\leq \epsilon \qquad\qquad (\alpha,\ \Delta_{\mathcal{V}}^{\text{suf}}(S^*, f, \mathbf{x}) \geq 0) \tag{17}$$

$$\implies \Delta_{\mathcal{V}}^{\text{nec}}(S^*, f, \mathbf{x}) \leq \frac{\epsilon}{1-\alpha}. \tag{18}$$

$\square$

#### A.1.2  PROOF OF LEMMA 4.2

**Lemma 4.2.** For $0 \leq \epsilon < \frac{\rho(f(\mathbf{x}), f_\emptyset(\mathbf{x}))}{2}$, denote $S^*_{\text{suf}}$ and $S^*_{\text{nec}}$ to be $\epsilon$-sufficient and $\epsilon$-necessary sets. Then, if $S^*_{\text{suf}}$ is $\epsilon$-super sufficient or $S^*_{\text{nec}}$ is $\epsilon$-super necessary,

$$S^*_{\text{suf}} \cap S^*_{\text{nec}} \neq \emptyset. \tag{19}$$

*Proof.* We will prove the result via contradiction. First recall that,

$$f_S(\mathbf{x}) = \mathop{\mathbb{E}}_{\mathbf{X}_{S^c} \sim \mathcal{V}_{S^c}} [f(\mathbf{x}_S, \mathbf{X}_{S^c})] \tag{20}$$

and, for any metric $\rho : \mathbb{R} \times \mathbb{R} \mapsto \mathbb{R}$,

$$\Delta_{\mathcal{V}}^{\text{suf}}(S, f, \mathbf{x}) \triangleq \rho(f(\mathbf{x}), f_S(\mathbf{x})) \tag{21}$$

$$\Delta_{\mathcal{V}}^{\text{nec}}(S, f, \mathbf{x}) \triangleq \rho(f_{S^c}(\mathbf{x}), f_\emptyset(\mathbf{x})). \tag{22}$$

Since $\rho$ is a metric on $\mathbb{R}$, it satisfies the triangle inequality. Thus, for $a, b, c \in \mathbb{R}$

$$\rho(a, c) \leq \rho(a, b) + \rho(b, c). \tag{23}$$

Now, let $S^*_{\text{suf}}$ be $\epsilon$-super sufficient and suppose

$$S^*_{\text{suf}} \cap S^*_{\text{nec}} = \emptyset. \tag{24}$$

This implies

$$S_{\mathsf{suf}}^* \subseteq (S_{\mathsf{nec}}^*)_c. \tag{25}$$

Subsequently, since $S_{\mathsf{suf}}^*$ is $\epsilon$-super sufficient,

$$\Delta_{\mathcal{V}}^{\mathsf{suf}}((S_{\mathsf{nec}}^*)_c, f, \mathbf{x}) \leq \epsilon. \tag{26}$$

As a result, observe

$$\rho(f(\mathbf{x}), f_\emptyset(\mathbf{x})) \leq \rho(f(\mathbf{x}), f_{(S_{\mathsf{nec}}^*)_c}(\mathbf{x})) + \rho(f_{(S_{\mathsf{nec}}^*)_c}(\mathbf{x}), f_\emptyset(\mathbf{x})) \qquad \text{triangle inequality} \tag{27}$$

$$= \Delta_{\mathcal{V}}^{\mathsf{suf}}((S_{\mathsf{nec}}^*)_c, f, \mathbf{x}) + \Delta_{\mathcal{V}}^{\mathsf{nec}}((S_{\mathsf{nec}}^*)_c, f, \mathbf{x}) \tag{28}$$

$$\leq \epsilon + \Delta_{\mathcal{V}}^{\mathsf{nec}}((S_{\mathsf{nec}}^*)_c, f, \mathbf{x}) \qquad S_{\mathsf{suf}}^* \text{ is } \epsilon\text{-super sufficient} \tag{29}$$

$$\leq 2\epsilon \qquad S_{\mathsf{nec}}^* \text{ is } \epsilon\text{-necessary} \tag{30}$$

$$\implies \epsilon \geq \frac{\rho(f(\mathbf{x}), f_\emptyset(\mathbf{x}))}{2} \tag{31}$$

which is a contradiction because $0 \leq \epsilon < \frac{\rho(f(\mathbf{x}), f_\emptyset(\mathbf{x}))}{2}$. Thus $S_{\mathsf{suf}}^* \cap S_{\mathsf{nec}}^* \neq \emptyset$. The proof of this result assuming $S_{\mathsf{nec}}^*$ is $\epsilon$-super necessary follows the same argument. $\qquad\square$

### A.1.3   PROOF OF THEOREM 4.1

**Theorem 4.1.** Let $\tau_1, \tau_2 > 0$ and $0 \leq \epsilon < \frac{1}{2} \cdot \rho(f(\mathbf{x}), f_\emptyset(\mathbf{x}))$. Denote $S_{\mathsf{suf}}^*$ and $S_{\mathsf{nec}}^*$ to be $\epsilon$-super sufficient and $\epsilon$-super necessary solutions to ($\mathsf{P_{suf}}$) and ($\mathsf{P_{nec}}$), respectively, such that $|S_{\mathsf{suf}}^*| = \tau_1$ and $|S_{\mathsf{nec}}^*| = \tau_2$. Then, there exists a set $S^*$ such that

$$\Delta_{\mathcal{V}}^{\mathsf{uni}}(S^*, f, \mathbf{x}, \alpha) \leq \epsilon \quad \text{and} \quad \max(\tau_1, \tau_2) \leq |S^*| < \tau_1 + \tau_2. \tag{32}$$

Furthermore, if $S_{\mathsf{suf}}^* \subseteq S_{\mathsf{nec}}^*$ or $S_{\mathsf{nec}}^* \subseteq S_{\mathsf{suf}}^*$. then $S^* = S_{\mathsf{nec}}^*$ or $S^* = S_{\mathsf{suf}}^*$, respectively.

*Proof.* Consider the set $S^* = S_{\mathsf{suf}}^* \cup S_{\mathsf{nec}}^*$. This set has the following properties:

(P1)  $S^*$ is $\epsilon$-sufficient because $S_{\mathsf{suf}}^*$ is $\epsilon$-super sufficient

(P2)  $S^*$ is $\epsilon$-necessary because $S_{\mathsf{suf}}^*$ is $\epsilon$-super necessary

(P3)  $|S^*| \geq \max(\tau_1, \tau_2)$ with $|S^*| = \tau_1$ when $S_{\mathsf{nec}}^* \subset S_{\mathsf{suf}}^*$ and with $|S^*| = \tau_2$ when $S_{\mathsf{suf}}^* \subset S_{\mathsf{nec}}^*$

(P4)  Via Lemma 4.1, we know $S_{\mathsf{suf}}^* \cap S_{\mathsf{nec}}^* \neq \emptyset$ thus $|S^*| < \tau_1 + \tau_2$

Then by (P1) and (P2)

$$\Delta_{\mathcal{V}}^{\mathsf{uni}}(S^*, f, \mathbf{x}, \alpha) = \alpha \cdot \Delta_{\mathcal{V}}^{\mathsf{suf}}(S^*, f, \mathbf{x}) + (1 - \alpha) \cdot \Delta_{\mathcal{V}}^{\mathsf{nec}}(S^*, f, \mathbf{x}) \tag{33}$$

$$\leq \alpha \cdot \epsilon + (1 - \alpha) \cdot \epsilon = \epsilon \tag{34}$$

and by (P3) and (P4) we have $\max(\tau_1, \tau_2) \leq |S^*| < \tau_1 + \tau_2$, $\qquad\square$

### A.1.4   PROOF OF COROLLARY 5.1

**Corollary 5.1.** Suppose for any $S \subseteq [d]$, $\mathcal{V}_S = p(\mathbf{X}_S \mid \mathbf{X}_{S^c} = \mathbf{x}_{S^c})$. Let $\alpha \in (0, 1)$, $\epsilon \geq 0$, and denote $\rho : \mathbb{R} \times \mathbb{R} \mapsto \mathbb{R}$ to be a metric on $\mathbb{R}$. Furthermore, for $f(\mathbf{X}) = \mathbb{E}[Y \mid \mathbf{X}]$ and $\tau > 0$, let $S^*$ be a solution to ($\mathsf{P_{uni}}$) such that $\Delta_{\mathcal{V}}^{\mathsf{uni}}(S, f, \mathbf{x}, \alpha) = \epsilon$. Then, $S^*$ satisfies the following conditional independence relations,

$$\rho\left(\mathbb{E}[Y \mid \mathbf{x}], \mathbb{E}[Y \mid \mathbf{X}_{S^*} = \mathbf{x}_{S^*}]\right) \leq \frac{\epsilon}{\alpha} \quad \text{and} \quad \rho\left(\mathbb{E}[Y \mid \mathbf{X}_{S_c^*} = \mathbf{x}_{S_c^*}], \mathbb{E}[Y]\right) \leq \frac{\epsilon}{1 - \alpha}. \tag{35}$$

*Proof.* All we need to show is that when $\mathcal{V}_S = p(\mathbf{X}_S \mid \mathbf{X}_{S^c} = \mathbf{x}_{S^c})$ and $f(\mathbf{X}) = \mathbb{E}[Y \mid \mathbf{X}]$, we have

$$f_S(\mathbf{x}) = \mathbb{E}[Y \mid \mathbf{X}_S = \mathbf{x}_S]. \tag{36}$$

Once this is proven, we can simply apply Lemma 4.1.

To this end, we have by assumption that $f(\mathbf{x}) = \mathbb{E}[Y \mid \mathbf{X} = \mathbf{x}]$ and, for any $S \subseteq [d]$, $\mathcal{V}_S = p(\mathbf{X}_S \mid \mathbf{X}_{S^c} = \mathbf{x}_{S^c})$. Then by definition

$$f_S(\mathbf{x}) = \mathbb{E}_{\mathcal{V}_{S^c}}[f(\mathbf{x}_S, \mathbf{X}_{S^c})] = \int_{\mathcal{X}} f(\mathbf{x}_S, \mathbf{X}_{S^c}) \cdot p(\mathbf{X}_{S^c} \mid \mathbf{X}_S = \mathbf{x}_S) \, d\mathbf{X}_{S^c} \tag{37}$$

$$= \int_{\mathcal{X}} \mathbb{E}[Y \mid \mathbf{X}_S = \mathbf{x}_S, \mathbf{X}_{S^c}] \cdot p(\mathbf{X}_{S^c} \mid \mathbf{X}_S = \mathbf{x}_S) \, d\mathbf{X}_{S^c} \tag{38}$$

$$= \int_{\mathcal{X}} \left( \int_{\mathcal{Y}} y \cdot p(y \mid \mathbf{X}_S = \mathbf{x}_S, \mathbf{X}_{S^c}) \, dy \right) \cdot p(\mathbf{X}_{S^c} \mid \mathbf{X}_S = \mathbf{x}_S) \, d\mathbf{X}_{S^c} \tag{39}$$

$$= \int_{\mathcal{Y}} y \left( \int_{\mathcal{X}} p(y, \mathbf{X}_{S^c} \mid \mathbf{X}_S = \mathbf{x}_S) \, d\mathbf{X}_{S^c} \right) dy \tag{40}$$

$$= \int_{\mathcal{Y}} y \cdot p(y \mid \mathbf{X}_S = \mathbf{x}_S) \, dy \tag{41}$$

$$= \mathbb{E}[Y \mid \mathbf{X}_S = \mathbf{x}_S]. \tag{42}$$

By applying Lemma 4.1, we have the desired result. $\qquad\square$

### A.1.5 PROOF OF THEOREM 5.1

**Theorem 5.1.** Consider an input $\mathbf{x}$ for which $f(\mathbf{x}) \neq f_\emptyset(\mathbf{x})$. Denote by $\Lambda_d = \{S, S^c\}$ the partition of $[d] = \{1, 2, \ldots, d\}$, and define the characteristic function to be $v(S) = -\rho(f(\mathbf{x}), f_S(\mathbf{x}))$. Then,

$$\phi_S^{\mathsf{shap}}(\Lambda_d, v) \geq \rho(f(\mathbf{x}), f_\emptyset(\mathbf{x})) - \Delta_{\mathcal{V}}^{\mathsf{uni}}(S, f, \mathbf{x}, \alpha). \tag{43}$$

*Proof.* Before we prove the result, recall the following properties of a metric $\rho$ in the reals:

(P1) $\forall a, b \in \mathbb{R}, \ \rho(a, b) = 0 \iff a = b$

(P2) for $a, b, c \in \mathbb{R}, \quad \rho(a, c) \leq \rho(a, b) + \rho(b, c).$

Now, for the partition $\Lambda_d = \{S, S^c\}$ of $[d] = \{1, 2, \ldots, d\}$ and characteristic function $v(S) = -\rho(f(\mathbf{x}), f_S(\mathbf{x}))$, $\phi_S^{\mathsf{shap}}(\Lambda_d, v)$ is defined as

$$\phi_S^{\mathsf{shap}}(\Lambda_d, v) = \frac{1}{2} \cdot [v(S \cup S^c) - v(S^c)] + \frac{1}{2} \cdot [v(S) - v(\emptyset)] \tag{44}$$

$$= \frac{1}{2} \cdot [\rho(f(\mathbf{x}), f_{S^c}(\mathbf{x})) - \rho(f(\mathbf{x}), f(\mathbf{x}))] + \frac{1}{2} \cdot [\rho(f(\mathbf{x}), f_\emptyset(\mathbf{x})) - \rho(f(\mathbf{x}), f_S(\mathbf{x}))] \tag{45}$$

$$= \frac{1}{2} \cdot [\rho(f(\mathbf{x}), f_{S^c}(\mathbf{x}))] + \frac{1}{2} \cdot [\rho(f(\mathbf{x}), f_\emptyset(\mathbf{x})) - \rho(f(\mathbf{x}), f_S(\mathbf{x}))] \qquad \text{by (P1)} \tag{46}$$

By (P2)

$$\rho(f(\mathbf{x}), f_\emptyset(\mathbf{x})) \leq \rho(f(\mathbf{x}), f_{S^c}(\mathbf{x})) + \rho(f_{S^c}(\mathbf{x}), f_\emptyset(\mathbf{x})) \tag{47}$$

$$\implies \rho(f(\mathbf{x}), f_{S^c}(\mathbf{x})) \geq \rho(f(\mathbf{x}), f_\emptyset(\mathbf{x})) - \rho(f_{S^c}(\mathbf{x}), f_\emptyset(\mathbf{x})). \tag{48}$$

Thus

$$\phi_S^{\mathsf{shap}}(\Lambda_d, v) = \frac{1}{2} \cdot [\rho(f(\mathbf{x}), f_{S^c}(\mathbf{x}))] + \frac{1}{2} \cdot [\rho(f(\mathbf{x}), f_\emptyset(\mathbf{x})) - \rho(f(\mathbf{x}), f_S(\mathbf{x}))] \tag{49}$$

$$\geq \frac{1}{2} \cdot [\rho(f(\mathbf{x}), f_\emptyset(\mathbf{x})) - \rho(f_{S^c}(\mathbf{x}), f_\emptyset(\mathbf{x}))] + \frac{1}{2} \cdot [\rho(f(\mathbf{x}), f_\emptyset(\mathbf{x})) - \rho(f(\mathbf{x}), f_S(\mathbf{x}))] \tag{50}$$

$$= \rho(f(\mathbf{x}), f_\emptyset(\mathbf{x})) - \Delta_{\mathcal{V}}^{\mathsf{uni}}(S, f, \mathbf{x}, \alpha). \tag{51}$$

$\square$

## A.2 ADDITIONAL EXPERIMENTAL DETAILS

In this section, we include further experimental details. All experiments were performed on a private cluster with 8 NVIDIA RTX A5000 with 24 GB of memory. All scripts were run on PyTorch `2.0.1`, Python `3.11.5`, and CUDA `12.2`.

### A.2.1 RSNA CT HEMORRHAGE

**Dataset Details.** The RSNA 2019 Brain CT Hemorrhage Challenge dataset (Flanders et al., 2020), contains 752803 images labeled by a panel of board-certified radiologists with the types of hemorrhage present (epidural, intraparenchymal, intraventricular, subarachnoid, subdural).

**Implementation.** Recall for this experiment, to identify sufficient and necessary masks $S$ for a sample $\mathbf{x}$, we considered the relaxed optimization problem (Fong et al., 2019; Kolek et al., 2022)

$$\arg \min_{S \subseteq [0,1]^d} \Delta_{\mathcal{V}}^{\mathsf{uni}}(S, f, \mathbf{x}, \alpha) + \lambda_1 \cdot ||S||_1 + \lambda_{\mathsf{TV}} \cdot ||S||_{TV}. \tag{52}$$

where $||S||_1$ and $||S||_{TV}$ are the $L^1$ and Total Variation norm of $S$, which promote sparsity and smoothness respectively and $\lambda_{\mathsf{Sp}}$ and $\lambda_{\mathsf{Sm}}$ are the associated. To solve this problem, a mask $S \in [0,1]^{512 \times 512}$ is initialized with entries $S_i \sim \mathcal{N}(0.5, \frac{1}{36})$. For 1000 iterations, the mask $S$ is iteratively updated to minimize

$$\alpha \cdot |f(\mathbf{x}) - f_S(\mathbf{x})| + (1 - \alpha) \cdot |f(\mathbf{x}) - f_S(\mathbf{x})| + \lambda_1 \cdot ||S||_1 + \lambda_{\mathsf{TV}} \cdot ||S||_{TV} \tag{53}$$

where for any $S$,

$$f_S(\mathbf{x}) = \frac{1}{K} \sum_{i=1}^{K} f((\tilde{\mathbf{X}}_S)_i) \quad \text{with} \quad (\tilde{\mathbf{X}}_S)_i = \mathbf{x} \circ \tilde{\mathbb{1}}_S + (1 - \tilde{\mathbb{1}}_S) \circ b_i. \tag{54}$$

Here the entries $(\tilde{\mathbb{1}}_S)_i \sim \text{Bernoulli}(S_i)$ and $b_i$ is the $i$th entry of a vector $\mathbf{b} = (b_1, \cdots, b_d) \sim \mathcal{V}$. In our implementation the reference distribution $\mathcal{V}$ is the unconditional mean image over the of training images and so $b_i$ is the simply the average value of the $i$th pixel over the training set. To allow for differentiation during optimization, we generate discrete samples $\tilde{\mathbb{1}}_S$ using the Gumbel-Softmax distribution. This methodology simply implies the entries $(\tilde{\mathbf{X}}_S)_i$ is a Bernoulli distribution with outcomes $\{b_i, x_i\}$, i.e. $(\tilde{\mathbf{X}}_S)_i$ is distributed as

$$\Pr[(\tilde{\mathbf{X}}_S)_i = x_i] = S_i \tag{55}$$

$$\Pr[(\tilde{\mathbf{X}}_S)_i = b_i] = 1 - S_i \tag{56}$$

For each $\alpha \in \{0, 0.5, 1\}$, during optimization we set $K = 10$, $\lambda_1 = 3$ and $\lambda_{\mathsf{TV}} = 20$ and use the Adam optimizer with default $\beta$-parameters of $\beta_1 = 0.9$, $\beta_2 = 0.99$ and a fixed learning rate of 0.01.

### A.2.2 CelebA-HQ

**Dataset Details.** We use a modified version of the CelebA-HQ dataset (Lee et al., 2020; Karras, 2017) which contains 30,000 celebrity faces resized to 256×256 pixels with several landmark locations and binary attributes (e.g., eyeglasses, bangs, smiling).

**Implementation.** Recall for this experiment, to generate sufficient or necessary masks $S$ for samples $\mathbf{x}$, we learn a model $g_\theta : \mathcal{X} \mapsto [0,1]^d$ via solving the following optimization problem:

$$\arg\min_{\theta \in \Theta} \mathbb{E}_{\mathbf{X} \sim \mathcal{D}_{\mathcal{X}}} \left[ \Delta_{\mathcal{V}}^{\mathsf{uni}}(g_\theta(\mathbf{X}), f, \mathbf{X}, \alpha) + \lambda_1 \cdot ||g_\theta(\mathbf{X})||_1 + \lambda_{\mathsf{TV}} \cdot ||g_\theta(\mathbf{X})||_{\mathsf{TV}} \right] \tag{57}$$

To learn sufficient and necessary explainer models, we solve Eq. (8) via empirical risk minimization for $\alpha \in \{0, 1\}$ respectively. Given $N$ samples $\{\mathbf{X}_i\}_{i=1}^N \overset{\text{i.i.d.}}{\sim} \mathcal{D}_X$, we solve

$$\frac{1}{N} \sum_{i=1}^N \left[ \Delta_{\mathcal{V}}^{\mathsf{uni}}(g_\theta(\mathbf{X}_i), f, \mathbf{X}_i, \alpha) + \lambda_1 \cdot ||g_\theta(\mathbf{X}_i)||_1 + \lambda_{\mathsf{TV}} \cdot ||g_\theta(\mathbf{X}_i)||_{\mathsf{TV}} \right]. \tag{58}$$

Here

$$\Delta_{\mathcal{V}}^{\mathsf{uni}}(g_\theta(\mathbf{x}_i), f, \mathbf{x}_i, \alpha) = \alpha \cdot |f(\mathbf{x}_i) - f_S(\mathbf{x}_i)| + (1 - \alpha) \cdot |f(\mathbf{x}_i) - f_S(\mathbf{x}_i)| \tag{59}$$

where is $f_S(\mathbf{x}_i)$ is evaluated in the same manner as in the RSNA experiment. For $\alpha = 0$, $\lambda_1 = 0.1$ and $\lambda_{\mathsf{TV}} = 100$. For $\alpha = 1$, $\lambda_1 = 1$ and $\lambda_{\mathsf{TV}} = 10$. For both $\alpha$, during optimization we use a batch size of 32, set $K = 10$ and use the Adam optimizer with default $\beta$-parameters of $\beta_1 = 0.9$, $\beta_2 = 0.99$ and a fixed learning rate of $1 \times 10^{-4}$

**Sampling.** To generate the samples in Figs. 4 and 5, samples we use the `CoPaint` method (Zhang et al., 2023). We utilize their code base and pretrained diffusion models with the exact the same parameters as reported in the paper to perform conditional generation. Everything used is available at https://github.com/UCSB-NLP-Chang/CoPaint.

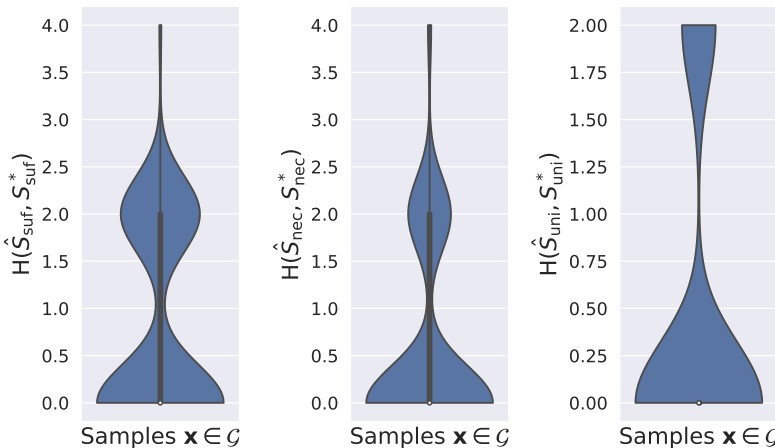

Figure 7: Hamming distances between computed and optimal solutions for $P_{\mathsf{suf}}$, $P_{\mathsf{nec}}$, and $P_{\mathsf{uni}}$

### A.3 ADDITIONAL EXPERIMENTS

#### A.3.1 SYNTHETIC EXAMPLE

We model features $\mathbf{X} \in \mathbb{R}^7$, where $X_i \sim \mathcal{N}(\mu_i, \sigma_i^2)$ for $i \in \{1, 4, 5, 6, 7\}$. The remaining features and response $Y$ follow:

$$X_2 = 2 \cdot X_1 + \epsilon, \quad Y = 4 \cdot X_2 \cdot \mathbf{1}_{\{X_2 > 10\}} + \epsilon, \quad X_3 = 4 \cdot Y + 15 \cdot X_4 \cdot \mathbf{1}_{\{X_4 > 0.5\}} + \epsilon \quad (60)$$

where $\epsilon \sim \mathcal{N}(0, 1)$. For $\mathbf{X} \in \mathcal{G} := \{\mathbf{X} \mid X_2 > 10, X_4 > \frac{1}{2}\}$, the data-generating process is represented by the directed acyclic graph (DAG) shown in Fig. 6 (note $X_5, X_6$ and $X_7$ are not depicted since they share no dependencies with any of the random variables). We can see that $Y \perp\!\!\!\perp \mathbf{X}_{\{1,5,6,7\}} | \mathbf{X}_{2,3,4}$ and $Y \perp\!\!\!\perp \mathbf{X}_{\{4,5,6,7\}}$. Thus, for $f(\mathbf{X}) = \mathbb{E}[Y \mid \mathbf{X}]$ and $\mathcal{V}_S = p(\mathbf{X}_{S^c} \mid \mathbf{x}_S)$, the solutions to $P_{\mathsf{suf}}$, $P_{\mathsf{nec}}$, and $P_{\mathsf{uni}}$ with $\tau = 4$ are:

$$S_{\mathsf{suf}}^* = \{2, 3, 4\}, \quad S_{\mathsf{nec}}^* = \{1, 2, 3\}, \quad S_{\mathsf{uni}}^* = \{1, 2, 3, 4\}.$$

In this experiment, we train a general predictor (a three-layer fully-connected neural network) to approximate $\mathbb{E}[Y \mid \mathbf{X}]$ and

1. Validate the sets listed above are the optimal solutions.

2. Demonstrate that common post-hoc interpretability methods struggle to recover these solutions.

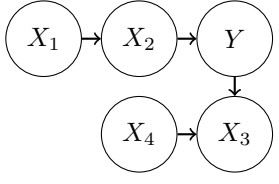

Figure 6: DAG modeling the data-generating process for $\mathbf{X} \in \mathcal{G}$

**Validation of Solutions.** For $\mathsf{type} \in \{\mathsf{suf}, \mathsf{nec}, \mathsf{uni}\}$, $\tau = 4$, and 100 samples $\mathbf{x} \in \mathcal{G}$ we compute solutions to $P_{\mathsf{type}}$, denoted as $\hat{S}_{\mathsf{type}}$, via exhaustive search. Fig. 7 shows that for all three problems, the Hamming distance between $\hat{S}_{\mathsf{type}}$ and $S_{\mathsf{ptype}}^*$ is equal to 0 for a majority of the samples in $\mathcal{G}$. These results indicate that the solutions computed via an exhaustive search do typically retrieve the correct solutions (the minor discrepancies are due to $f(\mathbf{X})$ being an approximation of $\mathbb{E}[Y \mid \mathbf{X}]$). More importantly, this setting is a clear example of how the unified approach provides a different perspective of importance. One would not be able to identify the set $S = \{1, 2, 3, 4\}$ as the most important one without directly solving the unified problem.

**Comparison with Post-hoc Methods** For our model $f$ and samples $\mathbf{x} \in \mathcal{G}$, we use Integrated Gradients, Gradient Shapley, Deeplift, and Lime to generate attribution scores. To identify whether these methods highlight sufficient and/or necessary features, and as done with our other experiments, we perform the following steps on the attribution scores for a sample $\mathbf{x}$ (so that the outputs of all methods are comparable)

1. We normalize the scores to the interval $[0, 1]$ via min/max normalization.

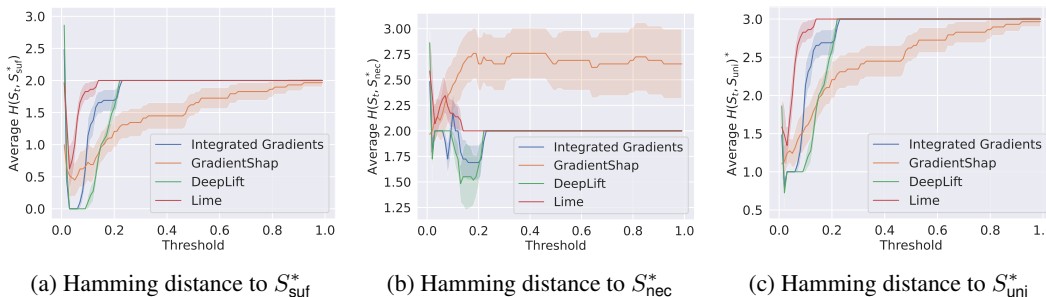

(a) Hamming distance to $S^*_{\text{suf}}$      (b) Hamming distance to $S^*_{\text{nec}}$      (c) Hamming distance to $S^*_{\text{uni}}$

Figure 8: Comparison of various post-hoc methods

2. We generate binary masks $S_t$ by thresholding the normalized scores with thresholds $t \in (0, 1)$

3. For $\text{type} \in \{\text{suf}, \text{nec}, \text{uni}\}$, we compute $H(S_t, S^*_{\text{type}})$, the Hamming distance between $S_t$ and the true solutions to $P_{\text{suf}}$, $P_{\text{nec}}$, and $P_{\text{uni}}$

The results in Fig. 8 illustrate that, in general, current post-hoc methods fail to recover the optimal sufficient, necessary, or unified solutions. For thresholds $t \in [0, 0.1]$, we see that Integrated Gradients and Deeplift recover solutions $S_t$ that match the optimal sufficient solution $S^*_{\text{suf}}$. This indicates these methods are capable of highlighting the sufficient features. Besides this observation, we see that for thresholds $t > 0.2$ and all three problems, nearly all methods recover solutions $S_t$ that have a Hamming distance $\geq 2$ to the optimal solution indicating that the solutions $S_t$ and optimal solutions $S^*$ differ by at least two elements. As a result, the conclusion is that most common methods do *not* detect sufficient solutions and *no* methods detect necessary or unified solutions.

## A.4 ADDITIONAL FIGURES

### A.4.1 RSNA CT HEMORRHAGE

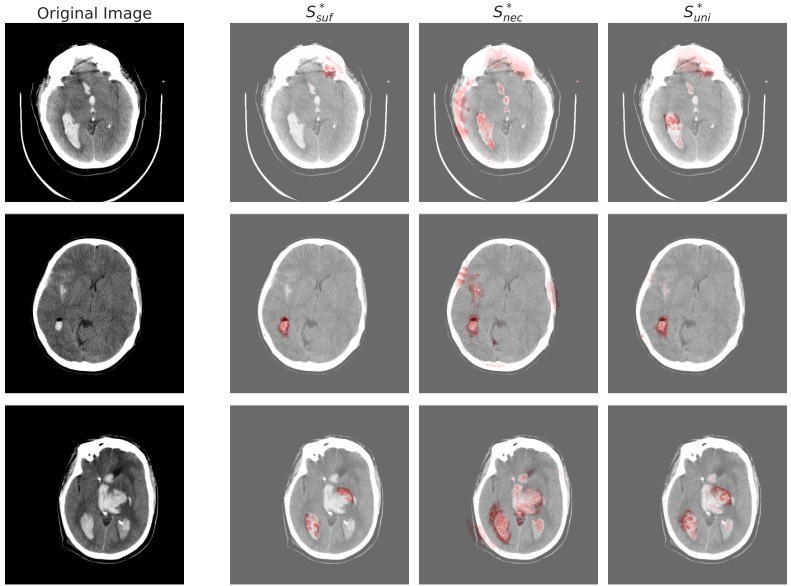

Figure 9: $S^*_{\text{suf}}$, $S^*_{\text{nec}}$ and $S^*_{\text{uni}}$ for various CT scans.

### A.4.2 CELEBA-HQ

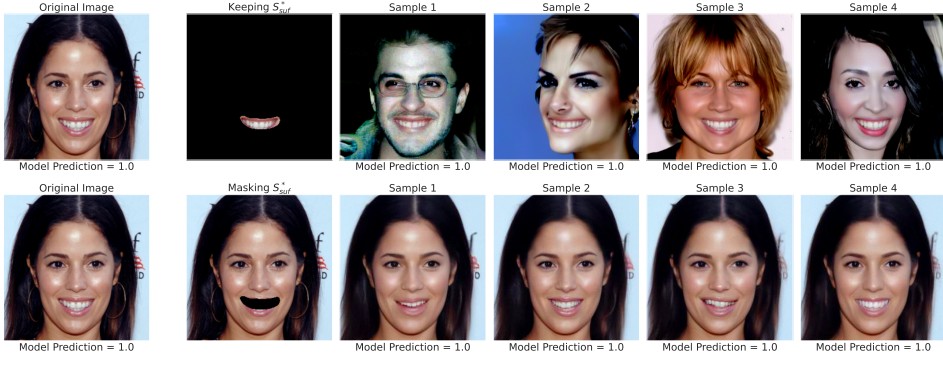

Figure 10: Images and model predictions by fixing and masking the sufficient subset $S^*_{\text{suf}}$

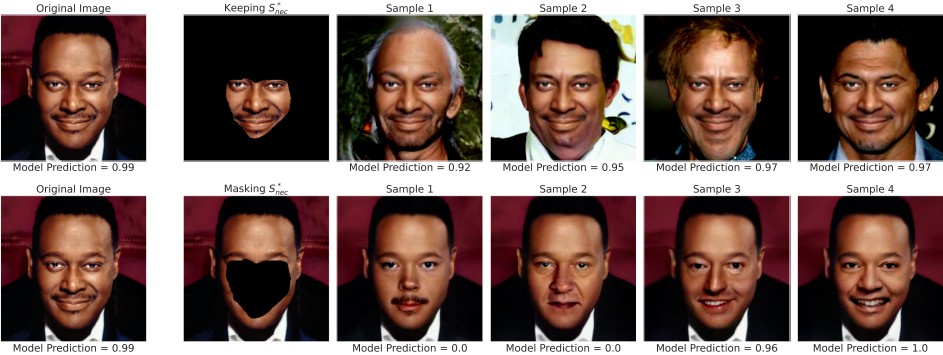

Figure 11: Images and model predictions by fixing and masking the necessary subset $S^*_{\text{nec}}$

