# OpenReview forum: "Sufficient and Necessary Explanations (and What Lies in Between)"
_ICLR.cc/2025/Conference — ICLR 2025 Conference Withdrawn Submission_

### Official Review · Reviewer_F3q7 · 2024-10-23

**Soundness:** 2
**Presentation:** 3
**Contribution:** 2
**Rating:** 5
**Confidence:** 4

**Summary:**

This paper proposes a novel measure to interpret predictions of black box machine learning models. Given a masking strategy for a model the authors define $\epsilon$-sufficient feature sets as sets whose masked prediction (mask= complement of S) deviates at most $\epsilon$ of the actual prediction with respect to some metric $\rho$ (deviation=$\Delta^{suf}$). In contrast, $\epsilon$-necessary feature sets are sets, where the masked prediction (mask=S) deviates at most $\epsilon$ from the fully masked prediction w.r.t $\rho$ (deviation=$\Delta^{nec}$). Given these notions, the authors aim to find the set $S$ with the smallest deviations. Instead of solving each problem individually, the authors propose to find a set that minimizes the weighted average $\Delta^{uni} := \alpha \Delta^{suf} + (1-\alpha) \Delta^{nec}$ with a hyper-parameter $0\leq \alpha \leq 1$, which they term "unified" problem, where the edge cases are the individual problems. The problem is further constrained, by finding a set with cardinality at most $\vert S \vert \leq \tau$. The main theoretical contribution is Theorem 4.1 that states the existence of $S$ with sufficiently small $\Delta^{uni}$, given sufficient and necessary solution and additional assumptions. Moreover, the solution satisfies conditional independence, if $f$ models the data-generating process (Corollary 5.1). Lastly, it is shown (Theorem 6.1) that the Shapley value of S treated as a joint player in the two-player game with S, [d]/S is bounded from below by $\Delta^{uni}$. In the experiments, the proposed is evaluated for different hyperparameter configurations on tabular data (6.1) and for image classification (6.2). For the restrictive setting in image classification, the authors propose an optimization objective that solves a relaxed version, which is specific to image classification, since exactly computing the objective is NP-hard. The authors conclude that the two proposed notions identify distinct aspects of predictions, which is showcased on examples for image classification.

**Strengths:**

1. The proposed necessary and sufficient explanations are an intriguing concept that complements existing approaches, such as the Shapley value that summarize "average contributions". Enriching the toolbox of interpretability methods with simple concepts is an important extension of existing work.
2. The paper is well-written, all theoretical claims are precisely stated and formally proven. The intuition behind the concepts are clear.
3. The proposed approximation method for image classification is interesting, but not sufficiently evaluated.
4. A first attempt to link the novel concepts to game-theoretic measures, such as the Shapley value, is promising.

**Weaknesses:**

1. **Limited discussion on computational aspects:** The paper introduces two interesting concepts, but the main limitation in practice is the optimization problem, which optimizes over all possible subsets (2^d), similar to the Shapley value. For the Shapley value, however, there exist efficient approximation techniques by evaluating the target for a collection of sampled subsets, which are unbiased and known to converge. In contrast, in Section 6.2 the authors propose approximation strategies for which no theoretical guarantees are given. The paper would highly benefit from such theoretical guarantees and a general approximation approach. Moreover, for the given applications, it should be carefully evaluated why these approximations yield appropriate solutions (see Q1 below).
2. **Limited link to other concepts:** Theorem 5.1 is a first step towards understanding connections to other methods, such as the Shapley value. However, this result is quite limited, since it considers a drastically reduced game of two players, instead of $d$ players, which does not give insights into the actual Shapley values that are usually computed for interpretability. The paper would strongly benefit from establishing such connections more carefully (see Q2 below), in particular given the claims made in the abstract.
3. **Comparison with existing attribution methods:** The authors claim that necessary and sufficient explanations yield more insights into explanations, which are undermined by some interesting examples in image classification. It remains unclear, how these methods compare to existing attribution methods. For instance, it would be helpful to formally state the difference between these concepts and carefully evaluate empirically which questions can be answered by the novel concept, which could not be targeted with existing methods. Moreover, some choices in the comparison are unclear to me (see Q3 below).
4. **Theorem 4.1:** The key assumption here is super-sufficient and super-necessary. I am not convinced that these properties are given in real-world examples (see Q4 below).
5. **Hyperparameters:** The method requires two hyperparameters $\alpha$ and $\tau$, which need to be chosen in advance, and have a high impact on the explanations, as shown by the experiments. It remains unclear how to choose these parameters in practice. It would be helpful to understand these choices better with sensible defaults.

**Questions:**

1. The proposed computation methods in Section 6.2. How well do they approximate the actual target? Could you do this analysis in a lower-dimensional setting, where ground-truths can be computed?
2. Do you have any insights on the method compared to the actual Shapley values of the model?
3. For the comparison of post-hoc explainability methods, what is the effect of the normalization? Is it negligible? Most methods already decompose the model's prediction, wouldn't it be sufficient to divide the attributions by the model's prediction to obtain normalized scores? Why don't you rely on the ranking of the attributions directly using $\tau$?
4. Is super-necessary and super-sufficient actually observed in practice? It seems counter-intuitive to me. For instance, for images, unmasking some parts of the image could lower the prediction significantly, since they could reveal a novel concept or another concept that contains the previously predicted concept.
5.  The purpose of the metric $\rho$ is unclear to me. It seems more intuitive to me to consider the model's output directly. What is the benefit of using the metric $\rho$? Why don't you use the model output directly, which is the common choice for local explanations? For necessary explanations, this also seems somewhat problematic: Consider a binary prediction, where the prediction is $\approx 1$, and the average prediction $f_\emptyset \approx 0.5$. Masking some features could substantially change the model's prediction to the opposite class, which would be captured with sufficient explanations. However, for necessary explanations, this would yield a similarly unsatisfying result (high $\Delta^{nec}), since the opposite class (prediction $\approx 0$) is equally far from the average prediction. Is this behavior intended? What is the reasoning behind it?
6. In Section 6.2. what are the resulting sizes of explanation sets that you find? Could you give some statistics on these for the datasets? Do you consider these sizes also in the comparison with post-hoc attribution methods? If so, in which way are they accounted for? E.g. I would expect to choose attributions of features such that the number matches with $\tau$?

**Minor**
- typo line 414, missing blank after "demonstrate"
- in Appendix A, the navigation displays "Proof of Theorem 6.1", while the section title is correctly referring to Theorem 5.1.

---

> ### Author Response · Authors · 2024-11-20
> **Response to reviewer**
>
> **Concern 1: Limited discussion of computational aspects**
>
> We thank the reviewer for the constructive comments. While it is true that the optimization problem is not tractable, we want to stress a couple of points:
>
> 1. Our contribution is not a tractable relaxation that identifies sufficient and/or necessary subsets. Instead, we provide precise and flexible notions of sufficiency and necessity, and bring to attention that these notions do not provide a complete picture of feature importance. To demonstrate our theoretical results in practice, we relied on computational methods that are already well-known in the literature [1,2,3,4].
>
> 2. Note that while the optimization problem may be intractable, the evaluation of $\Delta_{suf}, \Delta_{nec},$ and $\Delta_{uni}$ is not. Thus, we can still use these notions to evaluate how sufficient or necessary the explanations generated by different methods are. This allows us to provide a relative ranking of which method provides the best explanations in terms of sufficiency or necessity, which was not possible prior to our contribution.
>
> Your point is well-taken, however, and we will expand on the limitations of the computational aspects of solving the proposed problems in the revised manuscript.
>
> [1] "Interpretable Explanations of Black Boxes by Meaningful Perturbation," Ruth Fong, Andrea Vedaldi, ICCV 2017 [2] "Understanding deep networks via extremal perturbations and smooth masks," Ruth Fong, Mandela Patrick, Andrea Vedaldi, ICCV 2019 [3] "Cartoon explanations of image classifier," Stefan Kolek, et. al. ECCV 2022 [4] "Model Interpretability and Rationale Extraction by Input Mask Optimization," Marc Brinner, ACL 2023
>
> **Concern 2: Limited link to other concepts**
>
> First, we apologize if the claims in the abstract were misleading. Theorem 5.1 does establish a connection between the unified problem and the Shapley value, albeit for a two-player game. The reviewer is correct that is different from the way Shapley coefficients are typically used for approximating important features, but this does provide a connection between game-theoretic quantities and necessity/sufficiency methods which was unknown before. We do not find this different necessarily limiting, but rather a simpler and clearer picture of feature importance. By considering a subset $S \subseteq [d]$ and its complement $S^c$, we are simply differentiating the important features from those that are not.Moreover, note that the feature scores obtained with traditional methods (including via Shapley) are often (though not always) thresholded to differentiate between important and important features. Our formulation addresses this case.
>
> The comment is well-taken, and we will clarify these differences.
>
> **Concern 3: Comparison with existing attribution methods**
>
> We believe our empirical results do make a precise comparison of how different notions of importance differ. In particular:
>
> 1.  We show that necessary and sufficient need not be the same, but that they are related. In particular, we show in the two image classification settings that the sufficient explanations are subsets of the necessary ones.
>
> 2. We show that common post-hoc methods fail in retrieving necessary features, and most simply estimate (approximately) sufficient ones (see the RSNA and CelebAHQ experiments).
>
> 3. We show that our notions do allow for solutions that balance the trade-off between these notions of importance (portrayed in the tabular examples).
>
> **Concern 4: Theorem 4.1 assuming super sufficiency/necessity**
>
> Thank you for this comment. While super-sufficiency and super-necessity will not always hold in every scenario, they are not as strict as they seem. Super-sufficiency simply states that, given sufficient set (e.g. a region of an image), this set is super-sufficient if supersets of it are also sufficient. This indeed holds in the examples we provide: the brain CT scans and CelebAHQ. For the CT scans, once we fix a region with at least one hemorrhage, any larger region that contains this hemorrhage will be sufficient for predicting the label. Similarly in the CelebAHQ example, once ones fixes the region of the image that contains the smile, every superset of this region still allows the predictor to predict smile as expected. One can reason similar situations for the super-necessary settings: given a necessary set in the brain CT examples that contains all of the hemorrhages present in an image (i.e. a necessary set), any super-set of it will also be necessary.
>
> We hope these clarify our notions of super necessity and sufficiency, but we'd be happy to clarify further.

---

> ### Author Response · Authors · 2024-11-20
> **Response to reviewer (continued)**
>
> **Concern 5: Hyperparameters**
>
> Thanks for bringing this up. We stress that, unlike other problems where the hyperparameters are hard to interpret, both $\alpha$ and $\tau$ have precise meanings: the former controls the trade-off between sufficiency and necessity of the solution, whereas the latter determines the size of the reported features. As a result, there is no ``correct'' choice for these, but rather the choice should be determined by the specific problem domain or user preferences.
> For instance, if one is after a sufficient explanation, then $\alpha = 1$ is the correct choice -- correspondingly, $\alpha=0$ ensures a necessary one. We demonstrate, through theory and examples, that sufficient explanations may not capture necessary ones and vice versa. As a result, if one wishes to capture all such features, then $\alpha = 0.5$ is the appropriate choice.
>
> We will expand on this rationale in our revised version -- thanks for the comment!
>
> **Concern 6: Purpose of metric $\rho$**
>
> Thank you for this question, from which we identify two key components: one pertains to the choice of requiring $f_S(x)$ to be close to either $f(x)$ or $f_\emptyset$ (for sufficient or necessary features, resp.) instead of being close to 1 or 0 (analogous to the reviewer's suggestion of using ``using the model output directly''); the second pertains to the choice of the general metric $\rho$.
>
> For the former: we argue that measuring the distance of the expected perturbed function $f_S(x)$ with respect to the canonical choices of $f(x)$ and $f_\emptyset$ is more general. Consider the case of searching for sufficient features, and consider as an example, the case where $f(x) = 0.9$. Searching for a predictor $f_S(x)\approx 1$ (rather than one for which $f_S(x)\approx f(x)$ might provide features that are not sufficient for the predictor $f$ to produce the output $f(x)$ (only sufficient for producing $f_S(x)\approx 1$. This difference can be particularly important in cases where the predictions of $f(x)$ are calibrated, and thus their specific values carry specific meanings that we want to account for. Likewise, for necessary features, note that requiring $f_{S^c}(x)\approx 0$ instead of $f_{S^c}(x)\approx f_\emptyset$ (say, $\approx 0.5$) will result in a subset $S^c$ that is (on average) sufficient for predicting $f_{S^c}(x)\approx 0$ instead of necessary features for $f(x)$. This would result in features that are closer to counter-factual notions of explanations. Our notions, instead, simply require necessary features to be those without which the prediction is no better than a random guess.
>
> Lastly, we use a general metric $\rho$ to accommodate different prediction settings: a regression task, an $\ell_2$ norm might be appropriate, whereas in a multi-class classification settings the difference in maximum scores might be better suited, etc.
>
> **Concern 7: Questions about section 6.2**
>
> Thanks for bringing this up, as we realize that this could have been presented more clearly. As we comment on Sec. 6.2, and to ensure a consistent analysis, we first normalize all generated attribution scores to the interval $[0,1]$, and then obtain different important features by thresholding the scores at different values in $(0,1)$. Each threshold results in a generated explanation with a specific cardinality, or size (which is a monotonically decreasing function of the threshold). As can be seen in Fig. 2(a) and Fig(3), our results are provided as a function of this threshold (i.e. for all choices of this parameter), and we further include the sizes of the reported features by reporting $-\log(L^0)$, where $L^0$ refers to the (relative) cardinality of the reported features. As a result, this provides a complete picture that allows us to compare these methods for any value of $\tau$ (i.e. for any size of the reported important features).

---

> > ### Comment · Reviewer_F3q7 · 2024-11-26
> > **Follow-up**
> >
> > I thank the authors for answering my questions. I have some follow-up questions:
> > - **Comparisons**: Since the main contribution of your work is a novel concept for local explanations, I would highly appreciate to better understand the link to existing concepts, and theoretical differences. Specifically, since your novel concept is based on all possible perturbations/maskings, more extensive comparisons with the (actual) Shapley value (on feature level) as the leading concept of fair attributions would be beneficial. I think this would be best evaluated on a setting, where ground-truths can be computed, leaving the computational problem aside, e.g. evaluations across all choices of $\tau$, are the identified sets consistent? If not, what are the consequences? In my view, as of now, your experiments, while interesting, focus too much on the application side of the problem, where multiple factors (concepts, hyperparameters, approximation) interplay, which makes it hard to distinguish your proposed concepts of sufficiency and necessity from other factors.
> > - **super-sufficiency/necessity**: While I understand your intuition, do you have any empirical evidence that super-efficiency/necessity indeed holds in your setting or a dimensionality-reduced setting in real-world applications?
> > - **metric**: Looking at necessity, this would still imply that a set $S$ of a specific size $\tau$ with $f_{S^c}=0.45$ would be preferred over another set $T$ with $f_{T^c} = 0.6$, given $f_\emptyset = 0.5$. In other words, the set of necessary features $S$ for the target class 1 is actually a set of features that, if removed/masked, outputs more likely the opposite class 0 than any randomized prediction. Am I missing something? A simple solution could be the use the prediction instead, and cap this at $f_\emptyset$ and $f(x)$, which would also resolve the issues you mentioned on sufficiency.

---

> ### Author Response · Authors · 2024-11-27
> **Response to Reviewer**
>
> Thanks for the reply!  Here are our responses to your comments/concerns.
>
> **Comparisons**
>
> We have added an additional synthetic experiment where ground truths can be computed, and extensive comparisons can be made. The details are below.
>
> The experiment is the following:
>
> We model features $X \in \mathbb{R}^7$, where $X_i \sim \mathcal{N}(\mu_i, \sigma_i^2)$ for $i \in \set{1, 4, 5, 6, 7}$. The remaining features and response $Y$ follow:
>
> $$
> X_2 = 2\cdot X_1 + \epsilon
> $$
>
> $$
> Y = 4\cdot X_2 \cdot \mathbf{1}_{\{X_2 > 10\}} + \epsilon
> $$
>
> $$
> X_3 = 4\cdot Y + 15\cdot X_4 \cdot \mathbf{1}_{X_4 > 0.5} + \epsilon
> $$
>
> where $\epsilon \sim \mathcal{N}(0, 1)$. For $X \in \mathcal{G}:= \set{X \mid X_2 > 10, ~X_4 > 0.5}$, the data-generating process is represented by the directed acyclic graph (DAG) shown below
>
> $$
> X_1 -> X_2 -> Y -> X_3 <- X4
> $$
>
> with $X_5, X_6, X_7$ not connected to any variables. From the DAG, we can see that $Y \perp X_{\{1,5,6,7\}} | X_{2,3,4}$ and $Y \perp X_{\{4,5,6,7\}}$. Thus, for $f(X) = E[Y \mid X]$ and $V_{S} = p(X_{S^c} \mid x_S)$, the solutions to $P_{suf}$, $P_{nec}$, and $P_{uni}$ with $\tau = 4$ are:
>
> $$
> S_{suf}^* = \set{2,3,4}, ~~S_{nec}^* = \set{1,2,3}, ~~S_{uni}^* = \set{1,2,3,4}
> $$
>
> In this experiment, we train a general predictor (a three-layer fully-connected neural network) to approximate $E[Y \mid X]$ and 1) validate that the sets listed above are the optimal solutions, and 2) demonstrate that common post-hoc interpretability methods do not recover these any of these sets.
>
> Unfortunately, we cannot send figures through this forum, but we highlight the main takeaways from the experiments, and we’ll include the figures in the supplementary material.
>
> **Validating the solutions**
>
> For $type \in \set{suf, nec, uni}$, $\tau = 4$, and 100 samples $x \in \mathcal{G}$, we compute solutions, denoted as $\hat{S}_{type}$ to the sufficiency, necessity, and unified problem. We find that:
>
> 1) For $\approx$ 95% of the samples in $\mathcal{G}$, $\hat{S}_{suf} = \set{1,2,3}$, the solution to the sufficiency problem.
> 2) For $\approx$ 60% of the samples in $\mathcal{G}$, $\hat{S}_{nec} = \set{2,3,4}$, the solution to the necessity problem.
> 3) For $\approx$ 92% of the samples in $\mathcal{G}$, $\hat{S}_{uni} = \set{1,2,3,4}$, the solution the unified problem.
>
> These results indicate that the solutions computed via an exhaustive search do typically retrieve the correct solutions (the minor discrepancies are due to $f(X)$ being an approximation of $E[Y|X]$). More importantly, this setting is a clear example of when one would **not** be able to identify the set $S = \set{1,2,3,4}$ as the most important one unless you **directly** solve the unified problem.
>
> **Comparison with other methods**
>
> For our model $f$ and 100 samples $x \in \mathcal{G}$, we use Integrated Gradients, Gradient Shapley, DeepLift, and Lime to generate attribution scores. To identify whether these methods highlight sufficient and/or necessary features, and as done before in our manuscript, we perform the following steps on the attribution scores for a sample $x$ (so that the outputs of all methods are comparable)
>
> 1) We normalize the scores to the interval [0,1] via min/max normalization.
> 2) We generate binary masks $S_t$ by thresholding the normalized scores with thresholds $t \in (0,1)$.
> 3) For $type \in \set{suf, nec, uni}$, we compute $H(S_t, S^*_{type})$, the hamming distance between $S_t$ and the true solutions to $P_{suf}$, $P_{nec}$, and $P_{uni}$
>
> The main results from our analysis are the following:
>
> 1) There is no threshold in $t \in (0,1)$ for which **any** method recovers the true solution to $P_{suf}$,  $S_{suf}^* = \set{2,3,4}$. Furthermore, for $t > 0.1$ the average hamming distance, $H(S_t, S^*_{suf})$, is $ > 1$ for all methods, indicating that $S_t$ and $S_{suf}^*$ disagree by at least one element.
>
> 2) There is no threshold in $t \in (0,1)$ for which any method recovers the true solution to $P_{nec}$,  $S_{nec}^* = \set{1,2,3}$. In fact for $t > 0.6$, the average hamming distance $H(S_t, S^*_{nec})$, is $ > 2 $ for all methods, indicating that $S_t$ and $S_{suf}^*$ disagree by at least 2 elements.
>
> 3) For $t \approx 0.05$, integrated gradients and deeplift recover the true solution to $P_{uni}$, $S_{uni}^* = \set{1,2,3,4}$. However, for $t > 0.1$, the average hamming distance $H(S_t, S^*_{uni})$, is $ > 2 $ for all methods, indicating that $S_t$ and $S_{suf}^*$ disagree by at least 2 elements.
>
> We are in the middle of finishing this experiment, adding comparisons to the Shapley value, and updating the manuscript. We hope that the experiment and its results adequately address your concerns.
>
> **Metric**
>
> In the reviewer's example, set S is indeed more necessary than set T. We don't see any conflict between this and f_{S^c} < 0.5. Could the reviewer clarify why this is a concern?

---

> ### Author Response · Authors · 2024-11-27
> **Response continued**
>
> **Super-sufficiency/necessity**
>
> We provided weak evidence where super-sufficiency holds; e.g., any set that includes a brain hemorrhage in a CT scan will be super-sufficient for a 'good' predictor. However, we want to stress that these properties hold for a large class of data-generating processes. For example, they hold for distributions over $(X, Y)$ that can be modeled with a Markov Random Field. If the process can be expressed with a Markov Random Field, then the smallest $S_{suf}$ will be the parents of Y, and the smallest $S_{nec}$ will be all variables connected that have a path to $Y$. For example, consider
>
> X1 — X2 —Y — X3— X4
>
> X5     X6      X7
>
> Here $S_{suf} = \set{2,3}$ and $S_{nec} = \set{1,2,3,4}$. From this, it’s clear that adding any additional elements to $S_{suf}$ or $S_{nec}$ retains the sufficiency/necessity. Thus, super-sufficiency/necessity holds. Markov random fields appear in many real-world applications such as imaging, social network analysis, and genomics [1,2,3]. Thus, there are many real-world settings where super-sufficiency/necessity indeed holds.
>
> [1] Markov random field modeling in image analysis, Stan Z. Li Springer 2009
>
> [2] A network-specific Markov random field approach to community detection, Dongxiao He et al. AAAI 2018
>
> [3] A Markov random field model for network-based analysis of genomic data, Zhi Wei, Honghze Li, Bioinformatics 2007

---

> > ### Comment · Reviewer_F3q7 · 2024-11-28
> >
> > Thank you again for the additional comments and experiments.
> >
> > **Comparison**
> >
> > I thank the authors for the additional computational experiment. This seems to be a promising setup to confirm that your approximation works. For a comparison of methods, I would suggest rather to take a traditional local interpretability setup, maybe tabular data with few features, where exhaustive search is still feasible together with a fixed modeling choice for the conditional expectations that your approach and the competitors both use. To be more clear, what I would actually would like to better understand is **how** these concepts are different from the Shapley values, e.g. questions like: can we construct sufficient and necessary sets from the Shapley values? If not, why not? (there seems to be a strong discrepancy, why?) For $\tau=1$, is there a link to the TOP-1 Shapley value? If not, why not? A theoretical analysis of this would be highly beneficial, otherwise practitioners are left with two disconnected methods, one considering feature sets, and one individual attributions.
> >
> > **Minor comments regarding the experiment:** Please include a more formal description how you deduce the necessary and sufficient subsets. Regarding the comparison with Shapley values, I think the selection of TOP-k values is good for the sufficient features, I think for the necessary set, probably a different approach is better suited, since you would like to identify the set of Shapley values such that they sum to the baseline prediction, and for uni the combination of both may be better suited. Moreover, for a better comparison, it would be more meaningful to compare with the results from the exhaustive search rather than the actual sets, to eliminate the computational aspect.
> >
> > **super-efficiency/necessity**
> >
> > Thank you again for this suggestion. I think it is valuable to link to cases where this theoretically holds. I would be very interested, if this also holds in well-established benchmarks of local interpretability XAI literature. Moreover, if it does not hold, what are the consequences? I am still in doubt that this holds in practice and I think, as of now, your claim is not convincingly supported by empirical evidence.
> >
> >
> > **metric**
> >
> > I guess my concern here is that sufficient is depending on the class that is to be explained, whereas necessary (as a consequence of the example) is not. My understanding is now: Sufficient features are sufficient for the prediction of that class, whereas necessary features are necessary to make any prediction of any class (i.e. not predicting the baseline). I think there could be a problem arising from mixing these two into a single explanation, as effects could cancel.
> >
> >
> > Overall, I thank the authors for answering my questions and adding additional content. However, as of now, given the limitations of the current work, I decided to keep my score.

---

> > > ### Author Response · Authors · 2024-12-01
> > > **Response to reviewer**
> > >
> > > Thanks for the reply! Our responses to your concerns are below.
> > >
> > > **Sufficiency and necessity metric**
> > >
> > > We believe there is a misunderstanding here, and we are happy to clarify. Our notions of sufficiency/necessity are best understood when we take the model $f(X) = E[Y|X]$ and $V_S = p(X_S | x_{S^c})$. Then, for a sample $x$, we say a set $S$ is sufficient for the prediction if $|E[Y|x] - E[Y|x_s]| = 0$. In other words, $S$ is sufficient if, by only using the features in $S$, we can recover the original prediction that used all the features in $x$. On the other hand, our notion of necessity defines a set $S$ as necessary if, by removing $S$ and only using the complement, $S^c$, we have $E[Y | x_{S^c}] = E[Y]$. We say $S$ is necessary because we need $S$. Without it, our prediction is simply the naive prediction $E[Y]$, which does not use any information about $x$.
> > >
> > > We are not sure how any issues could arise from combining these notions and how cancellations may occur, as the reviewer implies.
> > >
> > > **Super sufficiency/necessity**
> > >
> > > Thank you for the suggestion, we will indeed add the analytical example we provided to the revised version of the paper. Furthermore, we will provide empirical evidence of super sufficiency/necessity holding in real-world examples.
> > >
> > > **More comprehensive synthetic experiment**
> > >
> > > We have written a global message to all the reviewers that details a more comprehensive synthetic experiment we believe addresses your concerns. Please take a look and let us know if you have any questions!

---

### Official Review · Reviewer_tCwB · 2024-10-29

**Soundness:** 4
**Presentation:** 2
**Contribution:** 3
**Rating:** 5
**Confidence:** 3

**Summary:**

The paper proposes to merge _sufficency_ and _necessity_ notions into one unified optimization problem $P_{\text{uni}}$ and solving this combined optimization problem. Therein, the paper is interesting and shows that this unified approach retrieves explanations unlike baseline methods.

**Strengths:**

- **Good Contribution**: While I think the contribution can be strengthened (see weaknesses), the presented approach is very interesting. Joining and optimizing both conditions leads to very interesting explanations, which seem to be not retrievable by other methods. The work is sound and carried out well.

- **Theoretical Connections to Game Theory**: The paper connects the novel optimization problem $P_{\text{uni}}$ with well-established concepts like the Shapley value. While I think the discussion and interpretation is under-developed (see weaknesses) this connection can have impact on the line of research regarding Shapely values and hierarchical explanations.

- **Synthetic Experiments**: I like that the paper studies the approach in synthetic environments! This is often not done anymore and can be very insightful of what the method is actually doing. The paper studies the stability of the method in this setting, which is good but leaves more to be desired.

- **Well written and structured**: The paper is clearly structured and reads well. Particularly, the related work section is very strong.

**Weaknesses:**

I think this paper **is borderline**: The paper feels a lot about the _what_ is being done, and not really about the _why_ this may be useful. The paper does not really motivate the use of $P_{\text{uni}}$ outside of the image domain and would benefit a lot from a wider evaluation in different domains such as language and tabular and the interesting insights that can be generated there.

- **Limited Contribution**: The contribution of this paper is rather incremental in nature. The notions of _sufficiency_ and _necessity_ are quite established concepts. Here they are just added together (with a linear combination) and jointly optimized for. While the paper does make interesting findings because of this combined lens (see strengths). It falls short in motivating and placing the contribution well into the XAI research. It is unclear when from an XAI perspective this joint approach is superior to other techniques. The conducted experiments seem rather exemplary than confirmatory in this regard. Only a small selection of created images are presented **without** examples for the baseline explanation methods this work compares against. The contribution could be strengthened by providing a broader comparison in different domains and actually showing how the explanations of other methods differ with the the new perspective.

- **Doubt about the stability**: The experiments in the work show that sufficiency and necessity optimized together can lead to interesting insights. However, a critical aspect of retrieving these explanations is by optimizing ($P_{\text{uni}}$) in some form. This is again intractable like solving ($P_{\text{suf}}$) and ($P_{\text{nec}}$). The authors acknowledge this fact (line 215, footnote 1). However, the authors point to two references where one is un-published work (Kolek et al. 2021), and other is a workshop article (Kolek et al. 2022). This makes me doubt that the retrieved solutions to ($P_{\text{uni}}$) are easily retrievable for various models and feature spaces. Neither the paper nor the appendix discusses this issue further. While Experiment 6.1 is concerned with this fact, stability is evaluated only on a synthetic or tabular example with low feature dimensionality. While the remainder of the paper deals with image explanations, which seems to be the main motivation for this work, the stability is left undiscussed.

- **Synthetic Setting is misused**: Given the above point stability is more interesting in higher-dimension settings (images, etc.). The synthetic environment would greatly benefit a _comparative study_ with other explanation methods. It would be interesting to see **how baselines fall short and be able to explain why** and then to show how the unification does not. This is in the current draft not done (also not in the exemplary setting of image explanations).

- **The Shapley-Perspective is Disconnected**: While I am glad about theoretical results linking this approach to game theoretic foundations (see strengths), Section 5.2 (connection of this method and Shapley) is quite disconnected to the rest of the paper. The results are presented technically but not put into context.

**Questions:**

- How stable are the retrieved explanations for the image case?

- What does it mean that _one searches for a player with a large lower bound_ in a two player game in practice. Can you put the results regarding the link of $P_{\text{uni}}$ into context. At the moment this connection is made and technically presented. It is not motivated or substantiated, why this is something good/bad or in-between.

---

> ### Author Response · Authors · 2024-11-20
> **Response to reviewer**
>
> **Concern 1: Limited contribution**
>
> We thank the reviewer for voicing their concerns. We believe this work does provide a meaningful contribution to the XAI community for the following reasons:
>
> -  While previous works have established certain notions of sufficiency and necessity, we introduce a definition of necessity that is different and--we argue--that better reflects what necessity should mean (see global comment that elaborates on this).
> - Importantly, we demonstrate that sufficient and necessary explanations need not be the same, but they are related (their intersection is not empty). This observation is what motivates the unified approach.
> - In all previous works, it remained unclear how exactly these notions of sufficiency and necessity relate to other notions of importance, such as conditional independence and Shapley values. We provide theoretical results that provide precise relations between these seemingly different perspectives
> - Lastly, through various experiments, we demonstrate how generating explanations along the necessity-sufficiency axis 1) allows us to detect important features that may otherwise be missed and 2) conclude that many post-hoc methods fall on the sufficiency side of this axis.
>
> We do apologize that these contributions were not clearly presented in the paper. In the revised version we will make these points clear.
>
> **Concern 2: Doubt about stability**
>
> We appreciate the comment. We agree that it is critical to optimize $P_{uni}$ in a manner that leads to efficient and stable solutions. The methods we employ are well-established in the literature [1,2,3,4] leading to solutions that are good and stable in practical settings spanning image and natural language settings. We apologize for not making this clear in the original version. We will address in the revised version by making this statement clearer in a new section before the experiments.
>
> [1] "Interpretable Explanations of Black Boxes by Meaningful Perturbation," Ruth Fong, Andrea Vedaldi, ICCV 2017 [2] "Understanding deep networks via extremal perturbations and smooth masks," Ruth Fong, Mandela Patrick, Andrea Vedaldi, ICCV 2019 [3] "Cartoon explanations of image classifier," Stefan Kolek, et. al. ECCV 2022 [4] "Model Interpretability and Rationale Extraction by Input Mask Optimization," Marc Brinner, ACL 2023
>
> **Concern 3: Synthetic setting is misused**
>
> Thanks for the comment. In the synthetic example, we focused on how the optimal feature set $S$ changes as a function of $\alpha$ with $\tau$ held constant, rather than comparing the solutions with other explanation methods (which do not depend on these parameters). Instead, we compare to other methods in more real and high-dimensional problems, including images. There, we indeed study how sufficient and necessary the ``important features'' provided by the different methods are by measuring their respective $\Delta_{suf}, \Delta_{nec}$ and $|S|$.
>
> **Concern 4: Shapley-perspective is disconnected**
>
> We are glad the reviewer enjoys the theoretical result of Theorem 5.2. The purpose of Theorem 5.2 is to motivate why minimizing $\Delta_{\uni}$ is a good idea at all, and we see our theoretical result as one such important justification. We agree with the reviewer that this was perhaps not clear enough, and we will make sure to stress it in revision.
>
> Regarding the implication and meaning of the result, consider the following.
>
> Denote $\Lambda_d = \{S, S^c \}$ the partition of $[d] = \{1, 2, \dots, d\}$ into two disjoint subsets, and define the characteristic function to be $v(S) = -\rho(f(x), f_{S}(x))$. Then, the following result holds.
>
> $$
> \phi^{shap}_S(\Lambda_d, v) \geq \rho(f(x), f_0(x)) - \Delta^{uni}_V(S, f, x, \alpha)
> $$
>
> Recall that a cooperative game is specified by a tuple $(\Lambda_d = \{S, S^c \}, v)$ and since $[d]$ can be partitioned into 2 sets $2^{d-1}$ ways, there are $2^{d-1}$ potential games. For every game, the Shapley value assigns an importance score to each of the two players (i.e. partitions) in a way that is fair (and satisfies other axiomatic properties). Note that the inequality above holds for all games; i.e. for all partitions of $[d]$. Thus, in solving for the $S$ with minimal $\Delta_{\text{uni}}$, one is identifying the game $(\Lambda_d = \{S, S^c \}, v)$ in which $S$ has the largest lower bound on its Shapley value. This result is interesting because it motivates minimizing $\Delta_{\text{uni}}$ through a game-theoretic interpretation: this is equivalent to selecting the game between players that maximize their difference of Shapley values.
>
> We hope this clarifies the result, and provide a stronger motivation to the story in manuscript. We will stress on this clarification in the revised manuscript!

---

> > ### Comment · Reviewer_tCwB · 2024-11-22
> > **Concerns still remain**
> >
> > Thank you to the authors for replying to my points. **Unfortunately, my concerns with and view of the paper still remain.**
> >
> > **Concern 4: Shapley result are not well substantiated/integrated**
> >
> > Yes, I get the technical side of the Shapley result, which is why I appreciate the result very much (c.f. strengths). However, the current paper or your elaboration on this topic in your reply does not go far enough. You write (bold added):
> >
> > > Thus, in solving for the $S$ with minimal $\Delta_{\text{uni}}$, one is identifying the game $(\Lambda_d = {S, S^c }, v)$ in which $S$ has the largest lower bound on its Shapley value. **This result is interesting** because it motivates minimizing $\Delta_{\text{uni}}$ through a game-theoretic interpretation: this is equivalent to selecting the game between players that maximize their difference of Shapley values.
> >
> > Yes! But **why** is this interesting? Your reply (and the additions in your updated manuscript) stops at this point where I would like to get some more intuition/guidance what novel things this theoretical results brings to the table apart from simply motivating your work by bringing in Shapley. What does _identifying the game $(\Lambda_d = {S, S^c }, v)$ in which $S$ has the largest lower bound on its Shapley value_ **really do**?
> >
> > **Concern 3: Missed opportunity with synthetic experiments**
> >
> > I see what you did with the _synthetic_ setting. However, my point still remains: The paper would have benefited a lot from actually comparing different XAI methods _that do not necessarily depend on your methods parameters_ in **synthetic** settings where you can control what a model is/should be doing. It would be very interesting to see how your proposal compares to established methods and weather it actually helps understanding something the established ones do not retrieve.
> >
> > > Instead, we compare to other methods in more real and high-dimensional problems, including images. There, we indeed study how sufficient and necessary the important features provided by the different methods are by measuring their respective and. There, we indeed study how sufficient and necessary the important features provided by the different methods are by measuring their respective $\Delta_{\text{suf}}$, $\Delta_{\text{nec}}$ and $\vert S\vert$.
> >
> > In this setting, you can't control for anything and we cannot really judge weather the results obtained by your method are substantially _better_ or _correct_ than the other methods. This discussion also touches my concern 1 (limited contribution), where I still do not see that the XAI community clearly learns when this unification is superior/preferred over other methods. However, I really think that this can be achieved by a proper validation in small-scale/synthetic experiments rather than high-dimensional settings.

---

> > > ### Author Response · Authors · 2024-11-26
> > > **Response to reviewer**
> > >
> > > Thanks for the quick response! Here are our responses to your comments/concerns.
> > >
> > > **Shapley result is not well substantiated/integrated**
> > >
> > > 1) The result motivates identifying a subset that is **equally** sufficient and necessary. Recall in our unified formula that $\alpha$ controls the sufficiency vs. necessity tradeoff. Our result, $\phi_{shap} > \rho(f, f_{\emptyset}) - \Delta_{uni}(S, f, x, \alpha)$, only holds for $\alpha = 1/2$, meaning that only under balanced sufficiency and necessity does the unified approach align with the Shapley value. Since the Shapley value is the only solution concept that is fair (i.e. satisfying the key properties of efficiency, symmetry, linearity, and null player), our result indicates that balancing sufficiency and necessity indirectly yields such favorable properties.
> > >
> > > 2) The result also motivates identifying a subset that is maximally **and** equally sufficient and necessary. In the XAI community, it is well regarded that features or sets of features with high Shapley values are more "important" since they provide larger contributions. Our result illustrates that, in a setting with 2 players, if one is after an important player (one with high Shapley value) then identifying one that is maximally sufficient and necessary (low $\Delta_{uni}) is a good strategy, as this player will surely have a large Shapley value. The result also implies that sets that are only sufficiency or necessity may not be as important (measured via the Shapley value) because optimizing for **only** sufficiency or necessity provides a smaller lower bound on the Shapley value.

---

> ### Author Response · Authors · 2024-11-26
> **Response continued**
>
> **Missed opportunity with synthetic experiments**
>
> To address this concern, we conducted a new synthetic experiment which we strongly believe highlights: 1) when/how solutions to the sufficiency, necessity, and unified problems differ, and 2) how current post-hoc methods fail to identify features that are sufficient and/or necessary.
>
> The experiment is the following:
>
> We model features $X \in \mathbb{R}^7$, where $X_i \sim \mathcal{N}(\mu_i, \sigma_i^2)$ for $i \in \set{1, 4, 5, 6, 7}$. The remaining features and response $Y$ follow:
>
> $$
> X_2 = 2\cdot X_1 + \epsilon
> $$
>
> $$
> Y = 4\cdot X_2 \cdot \mathbf{1}_{\{X_2 > 10\}} + \epsilon
> $$
>
> $$
> X_3 = 4\cdot Y + 15\cdot X_4 \cdot \mathbf{1}_{X_4 > 0.5} + \epsilon
> $$
>
> where $\epsilon \sim \mathcal{N}(0, 1)$. For $X \in \mathcal{G}:= \set{X \mid X_2 > 10, ~X_4 > 0.5}$, the data-generating process is represented by the directed acyclic graph (DAG) shown below
>
> $$
> X_1 -> X_2 -> Y -> X_3 <- X4
> $$
>
> with $X_5, X_6, X_7$ not connected to any variables. From the DAG, we can see that $Y \perp X_{\{1,5,6,7\}} | X_{2,3,4}$ and $Y \perp X_{\{4,5,6,7\}}$. Thus, for $f(X) = E[Y \mid X]$ and $V_{S} = p(X_{S^c} \mid x_S)$, the solutions to $P_{suf}$, $P_{nec}$, and $P_{uni}$ with $\tau = 4$ are:
>
> $$
> S_{suf}^* = \set{2,3,4}, ~~S_{nec}^* = \set{1,2,3}, ~~S_{uni}^* = \set{1,2,3,4}
> $$
>
> In this experiment, we train a general predictor (a three-layer fully-connected neural network) to approximate $E[Y \mid X]$ and 1) validate that the sets listed above are the optimal solutions, and 2) demonstrate that common post-hoc interpretability methods do not recover these any of these sets.
>
> Unfortunately, we cannot send figures through this forum but we highlight the main takeaways from the experiments, and we’ll include the figures in the supplementary material.
>
> **Validating the solutions**
>
> For $type \in \set{suf, nec, uni}$, $\tau = 4$, and 100 samples $x \in \mathcal{G}$, we compute solutions, denoted as $\hat{S}_{type}$ to the sufficiency, necessity, and unified problem. We find that:
>
> 1) For $\approx$ 95% of the samples in $\mathcal{G}$, $\hat{S}_{suf} = \set{1,2,3}$, the solution to the sufficiency problem.
> 2) For $\approx$ 60% of the samples in $\mathcal{G}$, $\hat{S}_{nec} = \set{2,3,4}$, the solution to the necessity problem.
> 3) For $\approx$ 92% of the samples in $\mathcal{G}$, $\hat{S}_{uni} = \set{1,2,3,4}$, the solution the unified problem.
>
> These results indicate that the solutions computed via an exhaustive search do typically retrieve the correct solutions (the minor discrepancies are due to $f(X)$ being an approximation of $E[Y|X]$). More importantly, this setting is a clear example of when one would **not** be able to identify the set $S = \set{1,2,3,4}$ as the most important one unless you **directly** solve the unified problem.
>
> **Comparison with other methods**
>
> For our model $f$ and 100 samples $x \in \mathcal{G}$ we use Integrated Gradients, Gradient Shapley, DeepLift, and Lime to generate attribution scores. To identify whether these methods highlight sufficient and/or necessary features, and as done before in our manuscript, we perform the following steps on the attribution scores for a sample $x$ (so that the outputs of all methods are comparable)
>
> 1) We normalize the scores to the interval [0,1] via min/max normalization.
> 2) We generate binary masks $S_t$ by thresholding the normalized scores with thresholds $t \in (0,1)$.
> 3) For $type \in \set{suf, nec, uni}$, we compute $H(S_t, S^*_{type})$, the hamming distance between $S_t$ and the true solutions to $P_{suf}$, $P_{nec}$, and $P_{uni}$
>
> The main results from our analysis are the following:
>
> 1) There is no threshold in $t \in (0,1)$ for which **any** method recovers the true solution to $P_{suf}$,  $S_{suf}^* = \set{2,3,4}$. Furthermore, for $t > 0.1$ the average hamming distance, $H(S_t, S^*_{suf})$, is $ > 1$ for all methods, indicating that $S_t$ and $S_{suf}^*$ disagree by at least one element.
>
> 2) There is no threshold in $t \in (0,1)$ for which any method recovers the true solution to $P_{nec}$,  $S_{nec}^* = \set{1,2,3}$. In fact for $t > 0.6$, the average hamming distance $H(S_t, S^*_{nec})$, is $ > 2 $ for all methods, indicating that $S_t$ and $S_{suf}^*$ disagree by at least 2 elements.
>
> 3) For $t \approx 0.05$, integrated gradients and deeplift recover the true solution to $P_{uni}$, $S_{uni}^* = \set{1,2,3,4}$. However, for $t > 0.1$, the average hamming distance $H(S_t, S^*_{uni})$, is $ > 2 $ for all methods, indicating that $S_t$ and $S_{suf}^*$ disagree by at least 2 elements.
>
> We are currently in the middle of updating the manuscript to include this experiment, but we hope that the experiment and the results adequately address your concerns. Thank you so much for engaging with us, and we look forward to hearing if this clarifies your concerns.

---

### Official Review · Reviewer_1XKc · 2024-11-03

**Soundness:** 3
**Presentation:** 3
**Contribution:** 1
**Rating:** 3
**Confidence:** 4

**Summary:**

The authors consider the problem of feature importance, i.e., quantifying the influence of different input features in the context of supervised learning, which has become quite popular in explainable AI in the recent past. They propose new measures of sufficiency and necessity of feature subsets, as well as a convex combination between the two. They also consider the optimisation problem of finding (small) sufficient/necessary feature subsets.

**Strengths:**

Interesting and up-to-date topic, well-written paper, thorough experimental study.

**Weaknesses:**

The first problem I have with this paper is that I find the definition of necessity flawed, or at least not very meaningful. The definition of sufficiency says that a feature subset S is sufficient if the function f projected to S remains eps-close to the original function using all features, which does make sense. Naturally, then, I would have expected that a subset S is called necessary if its complement is not sufficient. At least, this is the common duality between necessity and sufficiency/possibility also found in other branches of the literature (e.g., in modal/possibilistic logic).

Instead, the authors call a subset S necessary if f projected to the complement of S remains close to the default (average) prediction with no features. First, as already said, this does not establish a "duality", but apart from that, this definition is very questionable by itself. It somehow suggests that staying close to the default prediction is something bad, while moving away from it is good. Besides, it also leads to formal problems. For example, suppose that f is indeed a (close to) constant function. Then, according to the definition, all feature subsets are necessary, which is counter-intuitive.

The pathology outlined in the beginning of Section 4 is also a consequence of the flawed definition, I would say. Here, a feature subset is sufficient, but at the same time, its complement "contains important features". Again, this is completely counter-intuitive. How can a subset be sufficient if it misses important features?

Why is super-sufficiency an interesting property? Isn't it expected that adding more features will keep sufficiency? More interesting would be minimality: S is minimally sufficient if no feature can be removed from S without losing sufficiency. Analogously for super-necessity.

The "unification" is merely a convex combination of the two measures of sufficiency and necessity. Such convex combinations are routinely used in (multi-objective) optimisation, but why should we call them unification? Sure, one obtains both measures as special cases, for alpha=0 and alpha=1, respectively, but then it's more a generalisation than a unification.

The two perspectives in Section 5 are strange and in a sense again somewhat misleading. First, the notion of conditional independence is not defined in the standard way. Normally, conditional independence is a relation on random variables (used in probabilistic graphical models, for example). But why should we speak of conditional independence in the case of (5)?

The connection to the Shapley value is flawed as well. Normally, each feature is treated as a player and assigned a Shapley value. In Theorem 5.1, however, an entire feature subset S is treated as a single player, and its complement as a second player. Why should one consider such a partition as a game, and what does it help? A different game is then needed for every player. How do we connect this to the standard Shapley value? Eventually, with only two player S and S_c, the entire game is specified by the four values v(\emptyset),  v(S), v(S_c), v([d]). These are also the values/approximations looked at in the definition of sufficiency/necessity, so it's not very surprising to find a relationship here.

**Questions:**

See above.

---

> ### Author Response · Authors · 2024-11-20
> **Response to Reviewer**
>
> **Concern 1: Flawed definition of necessity**
>
> We appreciate the comments. If we understand correctly, the reviewer is saying that
> 1. Our notion of sufficiency is appropriate, i.e. $S$ is $\epsilon$- sufficient if $\rho(f(x), f_S(x)) \leq \epsilon$.
> 2.  Given this notion of sufficiency, necessity should be defined by saying that $S$ necessary if its complement is not sufficient, i.e $f(x) \neq f_{S^c}(x)$. Or, formalizing this a little, the reviewer's proposal is that $S$ is $\epsilon$-necessary if $\rho(f(x), f_{S^c}(x)) \geq 1-\epsilon$.
>
> First, we note that our definition of necessity implies the one suggested by the reviewer: if $\rho(f(x_{S^c}), f_{\emptyset}(x)) \leq \epsilon$ then $\rho(f(x), f(x_{S^c})) > 1-\tilde{\epsilon}$ where $\tilde{\epsilon} = 1 - (\rho(f(x), f_{\emptyset}(x)) - \epsilon)$. This is not surprising because if $f_{S^c}(x)$ is close to $f_{\emptyset}(x)$, then it is far from $f(x)$ as long as $f(x)$ and $f_{\emptyset}(x)$ are different.
>
> Second, and more broadly, the reason we define the sufficiency and necessity of $S$ using the quantities $\rho(f(x), f_S(x))$ and $\rho(f_{S^c}(x), f_{\emptyset})$ is because we want our notions to reflect the natural idea that, among all subsets $S \subseteq [d]$, the complete set $S = [d]$ should be the maximally sufficient and necessary subset. This is indeed the case with our definitions: for $S = [d]$, $\rho(f(x), f_S(x)) = \rho(f(x), f_(x)) = 0$ and $\rho(f_{S^c}(x), f_{\emptyset}(x)) = \rho(f_{\emptyset}(x), f_{\emptyset}(x)) = 0$ and so $[d]$ is $0$-sufficient and necessary as desired.
>
> We hope this clarifies why we define sufficiency and necessity in this specific way, and we'd be happy to elaborate further.
>
> **Concern 2: Flawed definition of sufficiency**
>
> Thank you for this comment. We respectfully disagree, and we believe that there is a simple example that demonstrates that there is no contradiction in having two disjoint minimal sufficient features: Consider a scenario where the label is determined by the presence of certain features (as it happens in the brain CT example in our experimental section, Sec. 7): the presence of any individual hemorrhage in the scan is sufficient to classify the scan as "positive". Assume there are two hemorrhages in the scan, $S_1$ and $S_2$, each of ``size'' of $|S_1| = |S_2| =k$ pixels (for simplicity of the argument). Then, any one of them individually is the smallest sufficient subset (of size $k$); i.e. $f_{S_1}(x) \approx f_{S_2}(x) \approx f(x)$. The same is true in many other settings where the presence of certain features determines the outcome of the task. Thus, in general, there is no reason why features in the complement of minimal sufficient ones can't provide useful information.
>
> We hope this clarifies the confusion, and we'd be happy to elaborate further.
>
> **Concern 3: super sufficiency/necessity**
>
> While not immediately obvious, one can verify that super-sufficiency does not always hold, as conditioning on a superset of a sufficient region can significantly alter predictions. For example, a predictor might classify a small fur region of a dog image as a bear, but will eventually change towards predicting the presence of a dog as the region grows and reveals other dog-like features.
>
> On the other hand, we indeed address the minimality of both sufficiency and necessity through the parameter $\tau$, which controls the size of feature subsets. Smaller $\tau$ emphasizes identifying the smallest sufficient and necessary sets.
>
> **Concern 4: Use of "unification"**
>
> This is valid point. We find the term "unified" appropriate because as the reviewer states, our formulation is a combination of sufficiency and necessity. Indeed, there's nothing particularly revolutionary about this convex combination of both objectives. We'd be happy to rename the resulting problem if the reviewer has a specific suggestion in mind!
>
> **Concern 5: Confusion with conditional independence perspective**
>
> Thanks for bringing up this point. The reviewer is correct that conditional independence is a relation on random variables. To be precise for $Y, X_{S},$ and $X_{S^c}$, $Y$ is independent of $X_{S^c}$ conditional on $X_S$ if, for all values of $Y, X_{S}, X_{S^c}$, we have $p(Y | X_{S},  X_{S^c}) = p(Y|X_{S})$. Informally, our result states that
> $$
> E[Y \mid x] \approx E[Y \mid x_{S^*}] \quad \text{and} \quad E[Y \mid x_{{S^c}^*}] \approx E[Y].
> $$
> Since this result pertains to conditional expectations for a fixed realization of $x$, this is a local conditional independence relation on means. However, it is still a conditional independence relation, albeit weaker than the standard notions. We apologize for loosely using the term conditional independence, which we will correct in the revised version.

---

> > ### Author Response · Authors · 2024-11-20
> > **Response to reviewer (continued)**
> >
> > **Concern 6: Confusion with Shapley perspective**
> >
> > We kindly disagree that the connection to the Shapley value is ``flawed''. The reviewer is correct in stating that traditionally every feature is treated as a player and Shapley values are computed for each feature. However, this is not necessarily "standard": in other work, see [1], Shapley values for sets of features are often defined, too. The motivation behind this is twofold. First, in most settings, a single feature barely contributes to a prediction (most notably, pixels in the case of images) and in reality, models often use sets of features (that may interact synergistically) to generate a prediction. Second, when considering sets of features as players, the computation of the Shapley value becomes tractable. Our result simply demonstrates that minimizing the unified approach is equivalent to identifying the two-player game for which one player has maximal Shapley value. While this result is simple and different from the traditional setting of $n$-players game, it still provides a precise motivation for speaking for both sufficiency and necessity via game-theoretic justifications, which has never been shown before.
> >
> > [1] "Feature importance: A closer look at shapley values and loco," Isabella Verdinelli, Larry Wasserman

---

### Official Review · Reviewer_LhWm · 2024-11-08

**Soundness:** 3
**Presentation:** 3
**Contribution:** 3
**Rating:** 6
**Confidence:** 4

**Summary:**

This paper introduces a novel approach to sufficient and necessary subset explanations, proposing a method to identify subsets along a (symmetric) spectrum of necessity and sufficiency. The authors present theoretical results illustrating the properties of this unified approach and its connections to established importance measures, such as conditional independence and Shapley values. They demonstrate the effectiveness of their approach through experiments on both tabular and image data.

**Strengths:**

- **Clarity and Novelty**: The paper is well-written and extends existing work with a creative new approach to subset explanations.
- **Theoretical Contributions**: The authors provide a thorough theoretical analysis of their method, detailing its properties and connections to existing notions of feature importance.
- **Empirical Evaluation**: The framework’s applicability is demonstrated through diverse experiments, supporting the practical relevance of the proposed subset explanations.

**Weaknesses:**

1. It would be helpful if the main paper, prior to the experiments, provided a more detailed overview of how solutions are computed.

2. In the experiments section, you mention using a relaxed optimization approach for image data. Does this imply that, for tabular data, the exact solution is found by examining all subsets? Additionally, is there any investigation into the guarantees or potential limitations of the relaxed approach compared to the original problem?

3. In the abstract (Line 15), "feature importance" is used to describe sufficient and necessary sets. However, "feature importance" may not be the most precise term for these concepts.

4. The implications of Theorem 5.1 require further elaboration. Comparing different subsets feels akin to comparing different players in different games or players competing in their own games. If I understand correctly, searching within P_{uni} involves finding the feature subset with the highest Shapley Value in its own game, evaluated against its complementary subset. This paragraph could benefit from further clarification to ensure readers fully grasp its significance.

**Questions:**

1. Why did you choose to use marginal distributions in defining sufficiency (L89)? Most literature on sufficient explanations relies on conditional distributions, and even in some experiments, you use conditional distributions.

2.  What advantage does controlling minimality with a parameter like \( \tau \) provide, rather than defining minimal sufficiency directly? Calibrating \( \tau \) could be complex, especially as the existence of solutions for specific \( \tau \) values is uncertain. If this choice were removed, how would it impact your approach?

3. Given the focus on local prediction, why not define necessity as an average out of a subset that diverges significantly from the current prediction of the considered observation (e.g., \(1 - \epsilon\)), instead of using the average prediction? This definition might align more closely with interpreting necessity as the minimal feature set required to maintain the current prediction of the considered observation.

4. In all your results, would it not be necessary first to assume the existence of a solution to the problem given \( \tau \)?

5. Your unifying solution is defined as a combination of sufficient and necessary subsets using weights \( \alpha \) and \( 1 - \alpha \). Why not allow for any convex combination where \( \alpha_1 + \alpha_2 = 1, alpha_1, alpha_2 in (0, 1) \)? Would that significantly impact your results?

### Missing Citation

In your related work on sufficiency, you may want to reference [1], a follow-up to Wang et al. (2021) published at NeurIPS 2022, which proposes a more tractable approach for tree-based models:

> [1]: "Consistent Sufficient Explanations and Minimal Local Rules for Explaining Any Classifier or Regressor," Salim I. Amoukou, Nicolas J.B Brunel, NeurIPS 2022.

I am willing to increase if my questions and weakness comments are addressed.

---

> ### Author Response · Authors · 2024-11-20
> **Response to Reviewer**
>
> **Concern 1: Detailed overview of how solutions are computed**
>
> We agree with the reviewer. In the revised manuscript, we will include a section before the experiment that details how approximate solutions are computed.
>
> **Concern 2: Solutions for tabular vs. image settings**
>
> To answer the first question, the reviewer is correct. In the tabular example, exact solutions were identified by examining all subsets of a fixed cardinality $\tau$. For the second question, the relaxed approaches we use are slight variations of methods introduced in [1,2,3,4]. None of these works provide any theoretical guarantees but the methods have been demonstrated to work well in practice. The main benefit of the relaxed approach is that it is tractable since it allows for the use of gradient based methods for optimization. We will definitely include a section before the experiments that comment on the these approaches and their benefits/limitations.
>
> [1] "Interpretable Explanations of Black Boxes by Meaningful Perturbation," Ruth Fong, Andrea Vedaldi, ICCV 2017
> [2] "Understanding deep networks via extremal perturbations and smooth masks," Ruth Fong, Mandela Patrick, Andrea Vedaldi, ICCV 2019
> [3] "Cartoon explanations of image classifier," Stefan Kolek, et. al. ECCV 2022
> [4] "Model Interpretability and Rationale Extraction by Input Mask Optimization," Marc Brinner, ACL 2023
>
>
> **Concern 3: Abstract line 15**
>
> Thank you for pointing this slight issue. We have reworded the abstract as follows:
>
> "To address this, we introduce and formalize two precise concepts—sufficiency and necessity—that characterize how sets of features contribute to the prediction of a general machine learning model."
>
> We think it is more useful to measure the sufficiency and necessity of sets of features (rather than the importance scores of individual features) because in most settings a single feature does not contributes to a prediction (most notably, in the case of images). In reality, models often use sets of features (that may interact synergistically) to generate a prediction. We hope the reviewer finds this edit more precise, which we have incorporated to our revised version.
>
> **Concern 4: Implications of Theorem 5.1**
>
> We appreciate the comment, and we will further elaborate on Theorem 5.1. In short, the reviewer is correct. Denote by $\Lambda_d =$ {$S, S^c$} the partition of $[d] = \{1, 2, \dots, d\}$, and define the characteristic function to be $v(S) = -\rho(f(x), f_{S}(x))$. Then, the following result holds.
>
> $$
> \phi^{shap}_S(\Lambda_d, v)\geq \rho(f(x), f_0(x)) - \Delta^{uni}_V(S, f, x, \alpha)
> $$
>
> A cooperative game is specified by a tuple $(\Lambda_d = \{S, S^c \}, v)$ and since $[d]$ can be partitioned into 2 sets $2^{d-1}$ ways, there are $2^{d-1}$ games. For every game, the Shapley value assigns an importance score to each of the two players in a way that is fair and satisfies other desirable properties. For each game, the above inequality holds. Thus, a clearer way to interpret the result is that, in solving for the $S$ with minimal $\Delta_{\text{uni}}$, one is identifying the game $(\Lambda_d, v)$ in which $S$ has the largest lower bound on its Shapley value. This result is interesting because it motivates minimizing $\Delta_{\text{uni}}$ through a game-theoretic interpretation by selecting the game with one player with the largest large Shapley value. We will stress on this clarification in the revised manuscript.
>
>
> **Concern 5: Notation in line 89**
>
> We thank the reviewer for pointing this out. This is actually a typo and we did not indent to use the term marginal distributions. What we intended to say is
>
> To define feature importance precisely, we use the average restricted prediction,
>
> $$
> f_S(x) = E_{X_{S^c} \sim V_{S^c}}[f(x_S, X_{S^c})]
> $$
>
> where $x_S$ is fixed, and $X_{S^c}$ is a random vector drawn from an arbitrary reference distribution $V_{S^c}$, that may/may not depend on $S^c$. For example, two commonly used reference distributions are the marginal distribution $V_{S^c} = p(X_{S^c})$ and conditional distribution $V_{S^c} = p(X_{S^c} \mid x_S)$.
>
> The revised manuscript incorporates this correction.
>
> **Concern 6: Advantage of $\tau$**
>
> First, note that some constraint on the cardinality of the subsets is needed -- otherwise, without it, the solution to the $\Delta^{uni}_V(S^*, f, x, \alpha)$ problem is achieved at $S^* = [d]$, i.e. using all features.
>
> Perhaps the reviewer meant to switch the objective of the optimization problem with the constraint on its size; i.e, posing a problem like $min_S |S|$ s.t $\Delta^{uni}_V(S^*, f, x, \alpha) \leq \epsilon$. This problem minimizes the cardinality of $S$ directly, but it has the equivalent difficulty of setting the appropriate parameter $\epsilon$. Lastly, one could consider $\epsilon =0 $ in the problem above, but this is not very useful for real settings since exact sufficiency (or necessity) might not be achieved.

---

> > ### Author Response · Authors · 2024-11-20
> > **Response to Reviewer (continued)**
> >
> > **Concern 6: Notion of necessity**
> >
> > We appreciate the question. We agree with the reviewer that the definition of necessity should align with the the idea that "a set $S$ is necessary if we cannot generate the original prediction without it, i.e. $f_{S^c}(x) \not\approx f(x)$." More formally, $S$ is necessary if $\rho(f(x), f_{S^c}(x)) \geq \Delta$ for some $\Delta > 0$. Note, while our definition seems to differ from this, it is in fact more general in that it implies this condition. If $\rho(f(x_{S^c}), f_{\emptyset}(x)) \leq \epsilon$ then $\rho(f(x), f(x_{S^c})) > \Delta$ where $\Delta = \rho(f(x), f_{\emptyset}(x)) - \epsilon$. Intuitively, if $f_{S^c}(x)$ is close to $f_{\emptyset}(x)$ then it is far from $f(x)$ as long as $f(x)$ and $f_{\emptyset}(x)$ are different.
> >
> > The reason we define necessity differently is because we want our notion of necessity to align with the intuitive principle that, "the set $S = [d]$ should be maximally sufficient and necessary for $f(x)$" With our definition, for $S = [d]$, we have $\Delta^{nec}(S, f, x) = \rho(f_{\emptyset}(x), f_{\emptyset}(x)) = 0$, indicating that $S = [d]$ is $0$-necessary (maximally necessary) as desired. In the revised version we further elaborate on this in Section 2 and provide a detailed comparison of our notion with classical definitions, along with its advantages, in the Appendix.
> >
> > **Concern 7: Existence of solution for given $\tau$**
> >
> > Recall the unified problem is to minimize $\Delta^{uni}_V(S, f, x, \alpha)$ subject to $|S| \leq \tau$. For any $\tau > 0$, we can always minimize $\Delta^{uni}_V(S, f, x, \alpha)$. Now, for very small $\tau$, the minimizer $S^*$ may not be a good one in the sense that $\Delta^{uni}_V(S, f, x, \alpha)$ may not be small, meaning $S^*$ is neither very sufficient nor necessary.
> >
> > If the reviewer is referring Lemma 4.1, Theorem 4.1 and Corollary 5.1, in these results we always do make a statement of the form "let $S^*$ be a solution for ...", and so here we are assuming we have a solution. Note here we are not assuming a solution exists (in fact it always does) but instead assuming we were able to solve for it. We hope this adequately addresses your question but, if not, please feel free to elaborate.
> >
> > **Concern 8: Any convex combination using $\alpha_1$ and $\alpha_2$**
> >
> > What the reviewer suggested is analogous to what we do: Note that you can define $\alpha_1 = \alpha$ and $\alpha_2 = 1-\alpha$. Thus, one can use these weights ($\alpha_1$ and $\alpha_2$) and they will always satisfy $\alpha_1 + \alpha_2 = 1$. Our definition just simplifies this process by using a single parameter that controls the trade-off.
> >
> > **Concern 9: Missing Citation**
> > We thank the reviewer for bringing this to our attention! We will include the new reference in the revised manuscript.

---

### Official Review · Reviewer_XZ9C · 2024-11-09

**Soundness:** 2
**Presentation:** 3
**Contribution:** 3
**Rating:** 5
**Confidence:** 4

**Summary:**

To address vague notions of feature importance in many XAI methods, the paper introduces formal definitions for sufficiency and necessity for local feature-based explanations. The authors further present a unified notion of sufficiency and necessity through a joint convex optimization problem to generate explanations that are both sufficient and necessary. The empirical results show that while existing feature importance methods can identify sufficient features, they fall short in finding necessary features.

**Strengths:**

The paper is mostly easy to read, every definition and theorem/results are relevant to the discussion and do not feel like the authors have included them for the sake of including math and notation. I also appreciate the authors for recognizing the need for auxiliary user studies in their limitations to highlight the fact that theoretical desiderata does not necessarily translate to real world impact/performance.

**Weaknesses:**

Having said that, the paper could improve on a couple of things:
1. Natural language explanation of "sufficiency" and "necessity" in Section 2 (I know this is in the introduction). Since they are the central concepts of the paper, it would be beneficial to really drive home what these mean in plain English for the reader.
2. The experiments section (detailed below)

The experiment setup was difficult to follow, a lot of missing details that I had to assume. For example, I couldn't find what the $L^0$ metric is defined (I am assuming that is the cardinality of the resulting $S$ for each method at thresholds). Similarly, the role of the threshold $t$ was unclear. I am assuming features with attribution/importance higher than $t$ will be included in $S$, but the paper does not explicitly mention that. It might be worth answering what the effect of adjusting $t$ is.

Moreover, motivation for the tabular dataset experiment is very weak. What is the significance of investigating stability (and a synthetic dataset)? I know in the introduction, the authors ask the question "when do necessary and sufficient sets differ"? Perhaps this is something that the authors should mention in this section too. Note that this is not the case for the image classification experiment, which explicitly outlines its purpose (line 346).

Most importantly, I, as a reader, am unsure of the main takeaway from the experiments apart from the technical contributions of the paper at work. For example, what should we make of the result that necessary explanations are sufficient? This is an important question to answer since the abstract mentions that the paper demonstrates how strictly sufficient and/or necessary explanations fall short in providing a complete picture.

**Questions:**

Questions:
- Line 89: Is $f(\mathbf{x}_{S}, \mathbf{X}_S)$  a typo? I'd imagine one of them should be from the complement?
- Theorem 4.1 shows existence of $S^{*}$, do you have any theoretical results on uniqueness? Or can there be a unique solution in the first place?
- What happens when $\rho(f(\mathbf{x}) - f_{\emptyset}(\mathbf{x})) < \varepsilon$? (i.e. prediction is close to baseline)

Notes:
- I would suggest following a more standard notation for set complements: $S^\complement$, though I know having a superscript in a subscript may not be ideal
- Figure references in Section 6.2.2 "Sufficiency vs. Necessity" paragraph may be incorrect?
- Line 414: "demonstrateLemma"
- Line 424: $g_{\theta}: \mathcal{X} \to \mathcal{X}$ is a really big typo. Only found out when I looked at A.2.

---

> ### Author Response · Authors · 2024-11-20
> **Response to Reviewer**
>
> **Concern 1: Natural language explanation of "sufficiency" and "necessity" in Section 2**
>
> We agree with the reviewer. We have reworded the first paragraph of section 2.1 to clarify the meanings of sufficiency and necessity with the following text:
>
> "We now present our proposed definitions of sufficiency and necessity. At a high level, these definitions were formalized to align with the following guiding principles:
>
> 1. $S$ is sufficient if it is enough to generate the original prediction, i.e. $f_S(x) \approx f(x)$.
> 2. $S$ is necessary if we cannot generate the original prediction without it, i.e. $f_{S^c}(x) \not\approx f(x)$.
> 3. The set $S = [d]$ should be maximally sufficient and necessary for $f(x)$.
>
> The principles P1 and P2 are natural and agree with the logical notions of sufficiency and necessity. Furthermore, because the full set of features provides all the information needed to make the prediction $f(x)$, it should thus be regarded as maximally sufficient and necessary (P3). With these principles laid out, we now formally define sufficiency and necessity."
>
> We hope this clarifies our definitions. We have also added further clarifications after the definitions of sufficiency and necessity in a revised version.
>
> **Concern 2: Experimental Setup**
>
> We apologize for the lack of clarity. 1) The $L^0$ refers to the relative cardinality of $S$ (|S|/d). 2) The threshold $t$ is used to generate the subsets $S$ and $S^c$. For a fixed $t$, all normalized importance scores higher than $t$ are included in $S$ and those lower than $t$ are not. As a result, $t$ controls the sensitivity of the choice of reporting important vs un-important features, for all methods.
>
> **Concern 3: Tabular Data Motivation**
>
> Thanks for pointing this out. Both tabular examples aim to demonstrate how optimal sets $S$ may change as we vary the levels of sufficiency and necessity we require. We argue this is an important question since, if they were very stable, one shouldn't be too concerned with the specific trade-off between sufficiently and necessity. In short, our experiments demonstrate that this is not the case: it is evident that as we demand higher levels of sufficiency (via increasing $\alpha$) the features in the optimal solution are constantly changing (measured via the Hamming distance to optimal necessary $S$).
>
> To make this point clearer, we have added a small description before the tabular examples that clearly outlines the objectives of these experiments. Thanks for the suggestion!
>
> **Concern 4: What are the experimental takeaways**
>
> This is a great point and we would be happy to explain. We believe the experimental results yield a few key important conclusions:
>
> 1. Sufficient and necessary sets differ: One and other notions of importance convey different observations about the response of a model (on sufficiency and necessity of the studied features, respectively). Our experiments demonstrate that, while one could have a situation where necessary and sufficient features coincide, in most common experimental settings and common post-hoc methods, these two notions differ.
>
> 2. There exist domains where the sufficient sets are subsets of the necessary sets (our image classification settings highlight this). As a result, a minimal sufficient explanation will not highlight necessary features, thus falling short in reporting all important features more broadly. Without our results (cite which one in particular) one would not be able to draw this conclusion.
>
> 3. Our definitions allow us to conclude that many current post-hoc methods identify small sufficient subsets, but not necessary sets. This finding is important as it allows us to better understand the limitations of current methodology (and to propose our unification strategy as a solution).
>
> **Concern 5: Theorem 4.1 Uniqueness:**
>
> We don't have a complete answer to the uniqueness of $S^*$, but we can provide a partial answer: Arguments for uniqueness are best understood when we consider the expected predictor under the true conditional distribution, i.e. $f(x) = E[Y \mid x]$ and $V_S= p(X_S \mid x_{Sc})$. From our results in section 5, a solution $S^*$ to the unified problem then satisfies:
>
> $$
> E[Y \mid x] \approx E[Y \mid x_{S^*}] \quad \text{and} \quad E[Y \mid x_{{S^c}^*}] \approx E[Y].
> $$
>
> If the joint distribution $p(X, Y)$ satisfies the Markov properties, then it is known that $S^*$ is unique [1]. However, this known result is strong because it is a global characterization (i.e. holds for all $x$). Nonetheless, we believe this is a direction worth exploring, and we will incorporate it into our manuscript.
>
> **Additional Question: Is $f(x_{S}, X_S)$ a typo? I'd imagine one of them should be from the complement?**
>
> Yes, the reviewer is correct. The equation should be $f_S(x) = E_{X_{S_c} \sim V_{S^c}}[f(x_S, X_{S^c})] $. Thanks for pointing this out!
>
> [1] Pearl, Judea. Causality: Models, Reasoning, and Inference. Cambridge University Press, 2009.

---

> > ### Author Response · Authors · 2024-11-20
> > **Response to Reviewer (Continued)**
> >
> > **Additional Question: What happens when $\rho(f(x), f_{\emptyset}(x)) < \epsilon$? (i.e. prediction is close to baseline)**
> >
> > We appreciate the reviewer asking this question. For simplicity, let $\epsilon = 0$. Then, if we have sample $x$ for which the prediction of the model $f$ is equal to the baseline prediction, this indicates that there is nothing particularly "interesting" about this $x$. More precisely, in these cases, the empty subset of features will be both sufficient and necessary. To see this, note that $S = \emptyset$ is the smallest sufficient subset because by assumption $\rho(f(x), f_{\emptyset}(x)) =0$ which implies $S = \emptyset$ is $0$-sufficient. Likewise, $S= \emptyset$ is the minimal necessary set because $f(x) = f_{\emptyset^{c}}(x)$, and so $\rho(f(x), f_{\emptyset}(x)) = \rho( f_{\emptyset^{c}}(x), f_{\emptyset}(\mathbf{x})) =0$ by assumption, which implies that $S = \emptyset$ is $0$-necessary. Thus, for $\rho(f(\mathbf{x}), f_{\emptyset}(\mathbf{x})) < \varepsilon$, we see that $S = \emptyset$ is approximately a good sufficient and necessary set, indicating that there are no "distinctive" features in $x$ that generate the prediction $f(x)$.
> >
> > We will incorporate this discussion to our manuscript.
> >
> > **Additional notes/small errors**
> >
> > We appreciate the reviewer for pointing out grammar and notion errors. We have corrected them in the revised version.

---

> > > ### Comment · Reviewer_XZ9C · 2024-12-02
> > >
> > > I thank the authors for their response! I still have two concerns:
> > >
> > > 1. Definition of Necessity
> > > 2. Main Takeaway of the Paper
> > >
> > > ### **Definition of Necessity**
> > > The reason I asked about what happens if the prediction is close to the baseline is that I thought the result did not make much sense. As the authors pointed out in their response, $\varnothing$ is the minimally necessary set according to the paper's definition. However, there also exist subsets of $[d]$ that are necessary as well (most notably $[d]$ itself).
> > >
> > > From the authors' response to Reviewer **1XKc**, it seems like $[d]$ being necessary is intentional. Why is this a "natural" idea? To me, it seems to go against the logical notion of **necessity** because it is possible to obtain the prediction without $[d]$.
> > >
> > > In addition, as noted in the updated manuscript, the definition of necessity is analogous to $\rho(f(\mathbf{x}), f_{S^c}(\mathbf{x})) \geq \Delta$ **if $f(\mathbf{x})$ and $f_{\varnothing}(\mathbf{x})$ differ**. Why is this a reasonable assumption?
> > >
> > > ### **Main Takeaway of the Paper**
> > >
> > > I understand that (1) sufficient and necessary sets differ, (2) there are domains where sufficient sets are subsets of necessary sets, and (3) existing explainability methods don't return necessary sets. I was hoping the authors would go beyond this and discuss the significance of these results.
> > > -  Why should we care that sufficient and necessary sets differ?
> > > -  What are the consequences of current explainability methods returning small sufficient sets? (i.e., what effect does this have on downstream tasks)
> > > -  Should we always look to return necessary sets? Is the answer domain-dependent?
> > >
> > > Unfortunately, there were no meaningful changes to Section 7 in the updated manuscript.

---

> > > > ### Author Response · Authors · 2024-12-03
> > > > **Response to Reviewer**
> > > >
> > > > Thank you for the reply! Our responses to your concerns are below:
> > > >
> > > > **Definition of Necessity**
> > > >
> > > > We think there is a misunderstanding here. In our response, we stated $\emptyset$ is the smallest necessary set when $\rho(f(x), f_{\emptyset}) = 0$. In general, $\emptyset$ is not the smallest necessary set.
> > > >
> > > > We also want to state that the complete set $[d]$ being necessary is not intentional. We strongly believe a notion of necessity should reflect that $[d]$ is not only necessary but maximally necessary. This is because, for a sample $x$ and model prediction $f(x)$, if we do not have all the feature values $x_1, .... x_d$, then it should **not** be possible to generate the original prediction value $f(x)$. If this were the case, then this implies $f_{\emptyset}(x) = f(x)$, which further means that $\emptyset$ is both sufficient and necessary.
> > > >
> > > > Lastly, regarding the statement that $f(x)$ and $f_{\emptyset}$ differ. We believe that one should be interested in generating sufficient and necessary explanations for $x$ such that these quantities differ. Otherwise, as stated earlier, $\emptyset$ is both sufficient and necessary, implying there are no distinctive features in $x$.
> > > >
> > > > **Main Takeaway of the Paper**
> > > >
> > > > Our paper's main takeaway is that sufficient and necessary explanations can often provide an incomplete picture of which features are important. They both have their utility, but one should not expect a sufficient set to be necessary and vice-versa. In turn, if one desires both, our unified approach provides such an explanation. The paper then provides a theoretical analysis of unified solutions along with different interpretations of sufficiency and necessity through notions of conditional independence and game theory. We then demonstrate how current methods often fall on the sufficiency side of the sufficiency-necessity axis.
> > > >
> > > > We believe our finding that current methods return small sufficient sets is important because it informs us as a community about what properties the explanations have. Our results highlight that many methods will highlight the small set of features that is enough to reconstruct the prediction and, in turn, may not highlight all the important features. Take, for example, our CT scan example. Our results show that most post-hoc methods will highlight a single hemorrhage but not all hemorrhages. Yet, we also identify that all hemorrhages are necessary for the prediction. This suggests that these common explainability methods look for the "smallest" or "simplest" explanation as opposed to the "complete explanation," which we believe is very interesting. This can be useful to individuals using these methods because they know the features the methods **do not** highlight are not necessarily unimportant.
> > > >
> > > > We believe that one should not always look for necessary sets. We think the choice of sufficiency vs. necessity is domain-specific, and both have their benefits. A simple example highlighting their utility is in loan approval. Suppose a bank uses a model to approve loans based on features like income, credit score, and employment history, and we identify that a high income and good credit score are sufficient for loan approval. This is useful as we can explain to applicants which factors guarantee loan approval, enabling better transparency. On the other hand, suppose we identify that a high credit score is necessary. This is equally important information to applicants because it informs them that, regardless of all the other details on the application, if the credit score is low, the applicant will never be approved for a loan. Here, a necessary explanation provides an actionable item for the applicant: they should first improve their credit score. Thus, a necessary explanation can also be useful. Overall, the choice between which one is more desirable is domain-specific. However, if you are unsure, then generating an explanation that is both, as our unified approach does, could be the safe course of action.
> > > >
> > > > Lastly, the changes to our paper, namely a new synthetic experiment, are located in the appendix and not directly in Section 7. We also have written a global note detailing this experiment for all reviewers to read. We apologize for not making this clear.

---

### Author Response · Authors · 2024-11-21
**Updates to the Manuscript**

We would like to thank all the reviewers for the insightful questions and comments about our work. We have uploaded a revised version of the manuscript that includes changes suggested by the reviewers (highlighted in blue). In short the major changes are the following.

1) We have rephrased parts of Section 2 to clearly motivate our proposed definitions of sufficiency and necessity. In this section we provide simple and intuitive "guiding principles" that motivated our definitions. We introduce the following text at the beginning of section 2.

> **Definitions** We now present our proposed definitions of sufficiency and necessity. At a high level, these definitions were formalized to align with the following guiding principles:
>
> P1.  $S$ is sufficient if it is enough to generate the original prediction, i.e. $f_S(x) \approx f(x)$.
>
> P2. $S$ is necessary if we cannot generate the original prediction without it, i.e. $f_{S^c}(x) \not\approx f(x)$.
>
> P3. The set $S = [d]$ should be maximally sufficient and necessary for $f(x)$.
>
> The principles P1 and P2 are natural and agree with the logical notions of sufficiency and necessity. Furthermore, because the full set of features provides all the information needed to make the prediction $f(x)$, it should thus be regarded as maximally sufficient and necessary (P3). With these principles laid out, we now formally define sufficiency and necessity.

2. The second change is we have included a short section before the experiments that details the methods we used to generate exact or approximate solutions to the sufficiency, necessity, and unified problems in the experimental section.

We believe these changes have greatly improved our work and we appreciate all the help! We encourage the reviewers to take a look and to let us know if they have any additional questions/concerns.

---

### Author Response · Authors · 2024-12-01
**Comprehensive Synthetic Experiment**

We have noticed that one of the main concerns raised by many reviewers was that the synthetic experiments were not comprehensive enough and/or lacked comparisons with traditional methods. To address this, we conducted the following experiment that we will add to the paper (Note: For simplicity, we use $s$, $n$, and $u$ to refer to $suf$, $nec$, and $uni$.)

The main findings are the following:

1) The optimal solutions to $P_{s}$, $P_{n}$, and $P_{u}$ for a fixed $\tau$ need not be the same.
2) The sets recovered by many common feature attribution methods are **not optimal solutions to the unified problem**

As a result, these findings highlight the utility of the unified framework because our approach can recover a small sufficient **and** necessary set, something other methods are not capable of.

The experiment is the following,

We model features $X \in \mathbb{R}^7$ where $X_i \sim \mathcal{N}(\mu_i, \sigma_i^2)$ for $i \in \set{1,4, 5, 6,7}$. The remaining features and response $Y$ follow

$$ X_2 = X_1 + \epsilon_1$$

$$ Y = X_2 + \epsilon_2 $$

$$ X_3 = 5\cdot Y + 5\cdot X_4 + \epsilon $$

where $\epsilon_i \sim \mathcal{N}(0,1)$. The **entire** data-generating process is represented by the directed acyclic graph (DAG) shown below

$$ X_1 -> X_2 -> Y -> X_3 <- X_4 $$

with the remaining features $X_5, X_6, X_7$ not connected to any other variables. In this setting, $Y \perp X_{\set{1,5,6,7}} | X_{\set{2,3,4}}$ and $Y \perp X_{\set{4,5,6,7}}$, thus for $\tau = 3$ solutions to $P_{s}$ and $P_{n}$ are

$$ S^*_{s} = \set{2,3,4}, ~~S^*_{n} = \set{1,2,3} $$

Furthermore, the distribution $(X, Y)$ is a multivariate normal. As a result, we can exactly compute $E[Y \mid X_S]$ for all $S \subseteq [d]$. With this setup, we solve $P_{s}$, $P_{n}$, and $P_{u}$ for $\tau = 3$ and accomplish the following

1) We validate that the optimal solutions for $P_{s}$ and $P_{n}$ are $S^*_{s}$ and $S^*_{n}$ respectively
2) Demonstrate that there is a subset of samples $\mathcal{X}_{g} \in \mathcal{X}$ for which $S^{*} = \set{1,3,4}$ minimizes the unified objective, $\Delta_u$, for $\alpha = 1/2$
3) We demonstrate that, for $x \in \mathcal{X}_{g}$, many common post-hoc methods do not identify $S^* = \set{1,3,4}$ as an important set.

**Validating solutions**

For a holdout set of 1,000 points, we perform an exhaustive search to generate solutions to $P_{s}$ and $P_{n}$, which we denote as $\hat{S}_s$ and $\hat{S}_n$ respectively.

For $P_{s}$, this entails calculating $\Delta_{s} = |E[Y|x] - E[Y|x_s]|$ for all $S \subseteq [d]$ and identifying the set for which this is minimal. Similarly, for $P_{n}$ we pick the $S$ that minimizes $\Delta_{n} = |E[Y|x_{S^c}] - E[Y]|$. Upon doing so, we identify that for all 1,000 points, $\hat{S}_s = S^*_s$ and $\hat{S}_n = S^*_n$ as expected

**Solution to unified problem**

For this holdout set, we also perform an exhaustive search to generate solutions to $P_{uni}$ for $\alpha \in \set{0, 0.25, 0.5, 0.75, 1}$. In doing so, we identify that, for $\alpha = 0.5$, there is a subpopulation of samples for which $S^* = \set{1,3,4}$ is nearly sufficient **and** necessary and accordingly, the optimal solution to $P_{u}$. We denote this subset of samples as $\mathcal{X}_g$.

---

> ### Author Response · Authors · 2024-12-01
> **Comprehensive Synthetic Experiment (continued)**
>
> **Comparison with post-hoc methods**
>
> Following the reviewers’ recommendation, for every $x$ in $X_g$, we use the following attribution methods to compute importance scores for every feature $i \in [d]$
>
> 1) Integrated Gradients
> 2) Gradient Shapley
> 3) Deep Lift
> 4) Lime
> 5) The Leave Out Covariate (LOCO) value, $|E[Y|x] - E[Y|x_{i^c}]|$
> 6) The Shapley value, $\phi^{shap}_i$ for 3 different contribution functions,
> 	- $v_1(S) = E[Y|x_s]$
> 	- $v_2(S) = - |E[Y|x] - E[Y|x_S]|$, the negative loss in information when using features in $S$
> 	- $v_3(S) = |E[Y|x_S] - E[Y]|$, the gain in information when using features in $S$
>
> We select the three features with the highest scores for each method to create a set $\hat{S}$. In doing so, we come to the following conclusions:
>
> **Conclusion 1**: For Integrated Gradients, Gradient Shapley, Deep Lift, Lime, and LOCO, for all $x \in X_g$, we have $\hat{S} = \set{2,3,4}$. In simple terms, all of these methods assign high scores to the features that comprise $S^*_{s} = \set{2,3,4}$. This implies the rankings of scores generated by these methods can be used to deduce the optimal sufficient set.
> Identifying why this is the case for most of these methods is a matter of future work. For LOCO, this is not very surprising. Since $E[Y|x] = E[Y|x_{S^*_{s}}]$, then for any $j$ not in $S^*_{s}$, we have
>
> $$
> |E[Y|x] - E[Y|x_{j^c}| = |E[Y|x] - E[Y|x_{S^*_{s}}, x_{j^c \setminus S^*_{s}}| = |E[Y|x] - E[Y|x_{S^*_{s}}| = 0.
> $$
>
> In other words, the LOCO parameter for all features not in $S^*_{s}$ will be 0. The remaining features, those in $S^*_{s}$, will have non-zero scores, and so selecting the features with the top scores is equivalent to identifying features that makeup $S^*_{s}$.
>
> **Conclusion 2**: Using Shapley values, for the samples in $X_g$, approximately 70% of samples have $\hat{S} = \set{1,2,3}$. For the other 30%, $\hat{S} = \set{2,3,4}, \set{1,2,3}$ or $\set{1,2,4}$ In other words, Shapley often assigns high scores to the features comprising $S^*_{u} = \set{1,3,4}$. In conclusion, the set created by picking the features with the highest Shapley values can often, but not always, be used to deduce the set that is a solution to the unified problem (for $\alpha=1/2$ ). This finding is interesting, as it suggests that combining information about how a feature $i$ contributes to all subsets $S \subseteq [d] \setminus \set{i}$, as the Shapley value does, is equivalent to measuring whether a feature is a member of a set that is both sufficient and necessary. Exploring why this happens is a matter of future work.

---

### Note · Authors · 2024-12-03

I have read and agree with the venue's withdrawal policy on behalf of myself and my co-authors.